



# Water Vapor climatologies in the extra-tropical Upper Troposphere and Lower Stratosphere derived from a Synthesis of Passenger and Research Aircraft Measurements

Patrick Konjari[1,4], Christian Rolf[1], Michaela I. Hegglin[1,5,6], Susanne Rohs[2], Yun Li[2], Andreas Zahn[3], Harald Bönisch[3], Martina Krämer[1,4], and Andreas Petzold[2,5]

[1]Forschungszentrum Jülich GmbH, Institute of Climate and Energy Systems 4 – Stratosphere, Jülich, Germany
[2]Forschungszentrum Jülich GmbH, Institute of Climate and Energy Systems 3 – Troposphere, Jülich, Germany
[3]Karlsruhe Institute of Technology, Institute of Meteorology and Climate Research, Karlsruhe, Germany
[4]Johannes Gutenberg-Universität Mainz, IPA, Mainz, Germany
[5]Bergische Universität Wuppertal, Institute for Atmospheric and Environmental Research, Wuppertal, Germany
[6]University of Reading, Department of Meteorology, Reading, UK

**Correspondence:** Patrick Konjari (p.konjari@fz-juelich.de)

**Abstract.**

Water vapor ($H_2O$) is a key trace gas in the upper troposphere (UT) and lowermost stratosphere (LMS), as it significantly influences the Earth's climate system through its roles in radiative forcing and cloud formation. However, accurate knowledge of the amount of $H_2O$ in this atmospheric region is still insufficient due to the difficulty and lack of precise in-situ and space-

borne measurements. This study presents a new methodology to derive adjusted $H_2O$ climatologies for the extra-tropical UT/LMS from regular measurements aboard passenger aircraft between 1994 and 2022 within the IAGOS (In-service Aircraft for a Global Observing System) research infrastructure. To this end, a synthesis of mean $H_2O$ is performed by sampling air mass bins of similar origin and thermodynamic conditions relative to the tropopause between a dataset from 60.000 flights applying the IAGOS-MOZAIC and -CORE compact hygrometer(ICH) and a data set of 500 flights using the more sophisticated IAGOS-

CARIBIC hygrometer. The analysis is, in combination with ECMWF ERA5 meteorological data, accomplished for the extra-tropical northern hemisphere, where the datasets have the largest common coverage. We find very good agreement in the UT, but a systematic positive humidity bias in the ICH measurements for the LMS. To account for this bias, mean $H_2O$ of the ICH are adjusted to the IAGOS-CARIBIC measurements based on a new mapping and adjustment approach. After applying this new method, the LMS $H_2O$ measurements are in good agreement between all investigated platforms. The extensive $H_2O$

data set from the compact IAGOS sensor can now be used to produce highly resolved $H_2O$ climatologies for the climatically sensitive LMS region.





## 1 Introduction

Over the past decades, upper tropospheric and lowermost stratospheric (UT/LMS) water vapor ($H_2O$) has gained increasing attention due to its significant impact on the global climate system (IPCC, 2023). Apart from the influence on ozone con-
centration (Kirk-Davidoff et al., 1999) and cirrus cloud formation, even small variations in UT/LMS $H_2O$ lead to substantial changes in radiative forcing (Riese et al., 2012; Banerjee et al., 2019; Gettelman et al., 2011). Radiative forcing calculations suggest that the increase in stratospheric $H_2O$ of 0.8 ppmv as derived from balloon soundings between 1980-2000 could have accounted for 30 % of the total anthropogenic forcing during that time period (Forster and Shine, 2002). On the other hand, satellite-borne measurements spanning back to the 1980s, have not shown significant long-term trends in stratospheric $H_2O$
over several decades (Hegglin et al., 2014; Konopka et al., 2022). For the future, however, most global climate models predict an increase in stratospheric $H_2O$ (e.g., Banerjee et al., 2019; Huang et al., 2020) and a corresponding surface warming of 0.2 – 0.3 W/m$^2$ per 1 Kelvin of global warming (Forster and Shine, 2002; Huang et al., 2020; Nowack et al., 2023). Nevertheless, there are still significant uncertainties in predicting the radiative forcing resulting from changes in stratospheric $H_2O$ (Huang et al., 2020). One of the reasons is that the low stratospheric $H_2O$ concentrations below 10 ppmv, and the even smaller changes
are difficult to detect with required statistical significance. Therefore, accurate detection of UT/LMS $H_2O$, as well as trends on a high temporal and spatial resolution are essential to better understand the role of $H_2O$ in this part of the atmosphere for the global climate system.

Global observation of $H_2O$ in the UT/LMS is provided by space-borne remote sensing instruments, such as the Microwave Limb Sounder (Hegglin et al., 2013). Due to their limited vertical resolution of several kilometers, however, the space-borne
observations cannot adequately resolve the high vertical gradients of $H_2O$ across the extra-tropical tropopause layer (Gettelman et al., 2011; Zahn et al., 2014). To address this limitation and provide UT/LMS $H_2O$ profiles with a high vertical resolution, airborne in-situ measurements play a crucial role. The IAGOS (In-service Aircraft for a Global Observing System; www.iagos.org) database offers $H_2O$ measurements from over 60.000 passenger aircraft flights, enabling to resolve the strong vertical and temporal $H_2O$ variations in the UT/LMS of the extra-tropical northern hemisphere (Zahn et al., 2014; Petzold
et al., 2020). For instance, based on IAGOS observations over North America, the North Atlantic, and Europe (40°-60°N each), Petzold et al. (2020) were able to resolve the pronounced seasonality of UT/LMS $H_2O$ in these regions with a vertical resolution of 0.3 km. This study revealed a near doubling of the $H_2O$ mixing ratio during summer compared to winter in the UT and the first kilometer above the thermal tropopause, with the strongest seasonality observed over the North Atlantic. The investigation of these seasonal and spatial UT/LMS $H_2O$ variations based on IAGOS measurements helps to quantify and
understand the processes that control the temporal and spatial variability of $H_2O$. These can be seasonal variations attributed to the Brewer-Dobson circulation and isentropic transport or short-term micro- to mesoscale $H_2O$ mixing between troposphere and stratosphere associated with turbulence and diabatic processes like overshooting convection (Gettelman et al., 2011). A better understanding of these processes helps to improve their representation in climate models. In this context, a large amount of $H_2O$ in-situ measurements as provided by IAGOS is of crucial importance to improve the accuracy of future $H_2O$ and
corresponding radiative forcing trends based on climate model predictions.



The IAGOS database consists of three datasets with instrument packages of different sorts: IAGOS-MOZAIC (MOZAIC: Measurement of Ozone by AIRBUS In-Service Aircraft; Marenco et al., 1998), IAGOS-CORE ((Petzold et al., 2015)) and IAGOS-CARIBIC (Dyroff et al., 2015). The MOZAIC project was the predecessor program to IAGOS-CORE, and the data were integrated into the IAGOS database afterwards. Combined, IAGOS-MOZAIC and -CORE contain ∼60.000 flights, both

providing humidity measurements by a compact humidity capacity sensor of same type (ICH; Neis et al. (2015a)). Compared to IAGOS-MOZAIC and -CORE, the IAGOS-CARIBIC dataset consists of a relatively small number of flights of ∼500. The IAGOS-CARIBIC package consists of sophisticated instruments, enabling high precision measurements of $H_2O$ even for very low stratospheric concentrations (Zahn et al., 2014). The ICH measurements, however, were found to loose precision for very dry stratospheric air masses (Rolf et al., 2023). In a previous intercomparison of IAGOS $H_2O$ observations (at that time taken

in the framework of the MOZAIC project), and research-aircraft based observations as part of the SPURT (Spurenstofftransport in der Tropopausenregion, trace gas transport in the tropopause region; Engel et al. (2006)) project, a lower $H_2O$ threshold for precise measurements of 10 ppmv was determined for the ICH at conditions typical of the extra-tropical LMS (Kunz et al., 2008). This lower detection limit for the ICH instruments was further constraint to 30 ppmv by means of a dedicated hygrometer intercomparison study (Rolf et al., 2023).

The aim of this study is to provide an improved dataset by mapping and adjusting the IAGOS-MOZAIC and -CORE measurements to observations from more precise instruments. Therefore, in order to quantify the $H_2O$ bias of IAGOS-MOZAIC and -CORE in the LMS, the first step of this study is to compare these data with the IAGOS-CARIBIC measurements that are able to resolve low LMS $H_2O$. Furthermore, sophisticated campaign measurements from 500 flights summarized in the JULIA (JüLich In-situ Airborne Data Base; Krämer et al., 2020) data base are included in the comparison.

The main challenge of this study is to devise an approach that allows for a valid intercomparison of the in-situ $H_2O$ datasets, despite the limited amount of IAGOS-CARIBIC and JULIA data, and the fact that the measurements of the datasets were performed on different platform at different times and in different regions. To address this challenge, we apply a robust new mapping methodology that enables an accurate comparison of $H_2O$ in air masses of similar atmospheric origin, thermodynamic conditions and seasons. Through this comparison, we can identify and quantify biases in the $H_2O$ measurements by

IAGOS-MOZAIC and -CORE in the LMS.

Based on the results of the intercomparison, we develop an adjustment methodology to the IAGOS-MOZAIC and -CORE $H_2O$ datasets in the LMS. This methodology allows us to account for the biases identified in the intercomparison, ensuring improved accuracy in the representation of the $H_2O$ variability in the northern mid-latitudinal UT/LMS region.

The paper is structured as follows: In Section 2, we provide a comprehensive overview of the datasets that are utilized in

the comparison presented in Section 3. Section 4 outlines the methodology employed for mapping and adjusting the IAGOS-MOZAIC and -CORE LMS $H_2O$ climatologies in dry LMS regions. In Section 5, we summarize the key findings of this study and provide an outlook on future research based on the adjusted IAGOS-MOZAIC and -CORE $H_2O$ climatologies.





## 2   H$_2$O datasets

### 2.1   Airborne in-situ H$_2$O measurements

Two sets of IAGOS H$_2$O data from about three decades of airborne in-situ measurements with a compact capacitive hygrometer
(IAGOS capacitive hygrometer: ICH; see Section 2.1.1), IAGOS-MOZAIC and IAGOS-CORE, are the basis of this study and
are combined into one dataset (IAGOS-MOZAIC&CORE). This extensive data set is validated against two others, IAGOS-
CARIBIC and JULIA, which are smaller, but have measured H$_2$O with more accurate instruments. The IAGOS-CARIBIC
water instrument applies two different sensor systems (WaSul and frost point hygrometer; see Section 2.1.2). The JULIA data

base primarily contains data from the high precision hygrometer FISH (Section 2.1.3). The datasets are summarized in Table 1.
A detailed intercomparison study of ICH, WaSul and FISH aboard a Learjet research aircraft was conducted as part of the
DENCHAR project and is described by Rolf et al. (2023).

### 2.1.1   H$_2$O from passenger aircraft: IAGOS - MOZAIC & CORE

The IAGOS-MOZAIC&CORE dataset (from now on just MOZAIC&CORE) (https://www.iagos.org/) provides measurements

of the relative humidity w.r.t. liquid (RH$_{liq}$), taken by the compact capacitive sensor ICH aboard commercial aircraft in the
period from 1995 to this day, and derive H$_2$O mixing ratios. The ICH relies on a water adsorbing dielectric film between two
parallel electrodes with its dielectric capacity depending on the relative humidity of the surrounding air. Measurements are
provided every 4 seconds, which corresponds to a horizontal resolution of about 1 km. The instruments are removed from the
aircraft every 3 months and are calibrated against a reference frost point hygrometer in an atmospheric simulation chamber

before being re-installed again. Details of the current instrument handling and data processing are described by (Petzold et al.,
2020).
The largest amount of the  60.000 flights in the dataset took place in the northern mid-latitudes between Europe and North
America (Figure 1a). Here, at flight altitudes of 9 - 13 km, heights of up to 5 km above the tropopause can be reached,
especially during the winter season on northern routes, while in the tropics only the upper troposphere is covered (Figure 1d).

### 2.1.2   H$_2$O from passenger aircraft: IAGOS-CARIBIC


The IAGOS-CARIBIC dataset (from now on just CARIBIC) provides measurements from more than 500 long-distance pas-
senger aircraft flights (Figure 1b and e) in the period from 1997 to 2020. The instruments installed in the CARIBIC packages
measure about 100 tracers and aerosol parameters (https://www.CARIBIC-atmospheric.com), including water vapor and total
(gaseous, liquid and ice phase) water.

Compared to MOZAIC&CORE, measurements are taken by more advanced instrumentation. The CARIBIC water instrument
applies two different sensor systems (measurement techniques), a modified (dual-channel PA-laser spectrometer) WaSul sensor
(for total and gasphase water) and a modified CR2 Buck frost-point hygrometer (FPH) sensor, with the FPH being used for a
regular in-flight calibration of the WaSul sensor. (Dyroff et al., 2015)





**Figure 1. Geographical distribution of H$_2$O measurements**. The plots indicate the flight density of IAGOS-MOZAIC&CORE, IAGOS-CARIBIC, and JULIA, both horizontally (a-c) and vertically (d-f). In (a-c), the flight density is given on a 5x5° grid for the IAGOS data sets, while for JULIA the single flight tracks and corresponding years are shown. Potential temperature is taken as vertical coordinate (d-f) with a resolution of 5K and a latitude resolution of 5°; the solid red lines represent the average thermal tropopause, calculated from ERA5 data.

### 2.1.3 H$_2$O from research aircraft: JULIA

The JUeLich In-situ Airborne Data Base (JULIA) contains H$_2$O data from more than 500 research aircraft flights during 46 campaigns that took place from 1994 onward. It contains precise measurements from advanced instrumentation of trace gases like H$_2$O, and cloud microphysical properties (Krämer et al., 2020). H$_2$O data are provided by instruments that have a high sensitivity to low stratospheric H$_2$O, with most measurements being performed by the Fast In-situ Stratospheric Hygrometer (FISH; Zöger et al., 1999), which is sensitive to low UT/LMS H$_2$O, with an accuracy of 6 - 8 % between 1 and 1000 ppmv



**Table 1.** Summary of the UT/LS airborne in-situ $H_2O$ data sets

| Product | Instruments | Measurement Quantity | Time period | Accuracy | Reference |
|---|---|---|---|---|---|
| IAGOS-MOZAIC&CORE | Capacitive Hygrometer (ICH) | $RH_{liq}$ | 1995 - 2021 | 4–7 % $RH_{liq}$ | Helten et al. (1998), Neis et al. (2015a, b) |
| IAGOS-CARIBIC | Photoacoustic laser spectrometer (WASUL), Frost-point hygrometer | $H_2O$ $T_{dew}$ | 1997 - 2020 | 4 % / 0.3 ppmv for $H_2O$=5ppmv | Dyroff et al. (2015) |
| JULIA | Lyman-$\alpha$ fluorescence hygrometer (FISH) | $H_2O$ | 1994-2021 | 7 % $\pm$ 0.3 ppmv | Zöger et al. (1999); Meyer et al. (2015) |

(Meyer et al., 2015). In contrast to the IAGOS flights, the research aircraft flights often reach deeper into the stratosphere and also cover the tropical LS, with altitudes of up to 20 km, which corresponds to potential temperatures of more than 500 K (Figure 1).

## 2.2 ECMWF ERA5 reanalysis data

The ERA5 reanalysis data (Hersbach et al., 2020) from the European Centre for Medium-Range Weather Forecasts (ECMWF)
provide meteorological parameters every hour in the period from 1959 onward. The ERA5 data are given on a 30 km horizontal resolution with 137 vertical model levels, reaching heights of up to 0.1 hPa. For this study, ERA5 data is used on a reduced resolution of 6 hours and 1 x 1°, but with the original vertical resolution. Along the flight paths of passenger (IAGOS) and research (JULIA) aircraft, the position relative to the first and, if present, second WMO thermal tropopause as well as the equivalent latitude derived from the potential vorticity fields are interpolated.

## 3 Intercomparison of airborne in-situ $H_2O$ datasets

In this section, a comparative analysis of the in-situ $H_2O$ products is performed, focusing on the UT/LMS region. Given the similar performance found in this study for the MOZAIC and CORE measurements, as detailed in Section 3.3, the primary emphasis is on the MOZAIC data, with the findings also accounting for CORE.
Figure 2a-c presents latitudinal cross-sections of $H_2O$ for the winter (December-February) season, using a resolution of 5° lat-
itude x 5 K potential temperature based on all available data from the different datasets. In the mid to high latitudes, potential temperature levels above 350 K predominantly correspond to air masses in the LMS during the winter season. Here, the mean





**Figure 2. Multi-annual latitudinal cross sections of $H_2O$ for the datasets IAGOS-MOZAIC&CORE, IAGOS-CARIBIC and JULIA.** (a-c) show the winter (December-February) mean $H_2O$ binned in 5° latitude x 5 K potential temperature; the solid red lines represent the average thermal tropopause, calculated from ERA5 data. For the winter season at 40 - 60°N, the probability density relative in coordinates relative to the thermal tropopause ($\Delta z$) and normalized per $\Delta z$ is shown for $H_2O$ (d-f) and $RH_{liq}$ (g-i) .

$H_2O$ values of MOZAIC are significantly higher (10-20 ppmv) than the values reported by CARIBIC and JULIA (5-10 ppmv). The $H_2O$ frequency distribution relative to the thermal tropopause (TTP, altitude bin width: 0.25 km) shown in Figure 2d-f for the winter 40 - 60°N domain underpins the noticeable contrasts between the three datasets. Specifically, for MOZAIC, a consistent moist bias is evident from heights of 1 km above the TTP. This bias is becoming more pronounced at higher altitudes. Additionally, MOZAIC shows a more pronounced $H_2O$ variability compared to CARIBIC and JULIA at heights of 1 km above the TTP. This behavior is closely linked to the measurement quantity $RH_{liq}$ of the compact IAGOS humidity sensor




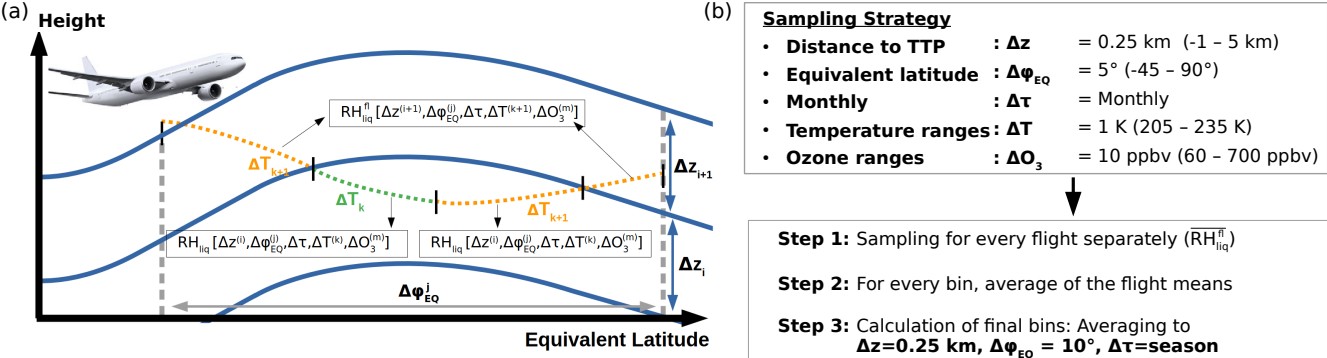

**Figure 3. Schematic illustration of the data sampling strategy to compare the different $H_2O$ products**. (a) The blue lines indicate constant height levels relative to the tropopause, and the two dashed grey lines mark a 5° equivalent latitude bin; see top box of (b) for the definition of the sampling bins of distance to the thermal tropopause ($\Delta z$), equivalent latitude ($\Delta\varphi_{EQ}$), time ($\Delta\tau$), temperature ($\Delta T$), and ozone ($\Delta O_3$). The bottom box of (b) lists the different steps to derive the final sampling bins used for the intercomparison of the $H_2O$ datasets.

ICH (see Section 3.3). The $RH_{liq}$ data from CARIBIC and JULIA indicate that $RH_{liq}$ values frequently drop below 10 % at these altitudes, and they are often below 5 % at altitudes of 3 km and higher than the TTP (Figure 2h-i). In contrast, the $RH_{liq}$

profile of MOZAIC displays a less strong decrease of $RH_{liq}$ in the LMS and generally higher values compared to CARIBIC and JULIA.

As already mentioned in the Introduction, previous research campaign flights where the compact ICH sensor was operated along with the sophisticated $H_2O$ instrument FISH revealed that the sensitivity of the ICH sensor decreases significantly for very low $RH_{liq}$ values below 10 % (Neis et al., 2015a, b). The reason for this loss of sensitivity of the ICH sensor in the LMS

is attributed to the adiabatic compression effect. As the air flows into the inlet towards the sensor, it undergoes heating in the range of 20 to 30 K. Consequently, even though the stratospheric humidity values are already very low, the measured values at the sensor decrease by the factor of 10, resulting in a sensitivity loss of the ICH for these very small relative humidity values. More details about this effect are discussed in (Petzold et al., 2020).

When comparing the $H_2O$ distributions shown in Figure 2e-f between JULIA and CARIBIC, both originating from sensors

sensitive to low stratospheric $H_2O$ values, some differences are nevertheless noticeable. First, the spatial and seasonal coverage of JULIA and CARIBIC datasets differs, contrary to the very good temporal and spatial agreement between CARIBIC and MOZAIC&CORE (Figure 1). Moreover, the research campaign measurements in JULIA were often conducted under specific atmospheric conditions, such as during studies of troposphere-to-stratosphere exchange processes. As a result, cases of anomalous $H_2O$ concentrations may be over-represented in the JULIA data set. This is evident, for instance, in a frequency of

occurrence of $H_2O$ between 20 and 50 ppmv at distances of 3 km and more above the thermal tropopause (TTP) in Figure 2b, where the mean winter extratropical $H_2O$ values are expected to be below 10 ppmv on average (Zahn et al., 2014). However, at higher altitudes, corresponding to potential temperatures above 400 K, JULIA provides a climatological perspective of $H_2O$





from the tropics to the northern sub-polar regions (Figure 2a). In these potential temperature ranges, $H_2O$ concentrations exhibit seasonal variations due to the Brewer-Dobson circulation, while the effects of short-term UT-to-LMS mixing processes

do not significantly contribute to the $H_2O$ distribution on a short time scale in the order of days (Gettelman et al., 2011).

As discussed in this section, MOZAIC&CORE overestimate low $H_2O$ in most parts of the extratropical LMS due to sensor limitations in capturing low values of $RH_{liq}$ values that are common in this part of the atmosphere. To further quantify the MOZAIC&CORE bias in the LMS, we developed an air mass mapping approach that allows for a better intercomparison with the CARIBIC $H_2O$ data. This method will be detailed in the next section.

## 170    3.1    Mapping approach to compare in-situ $H_2O$ datasets

A mapping approach is used as the method of evaluation of MOZAIC&CORE on the basis of CARIBIC, focusing on the primary MOZAIC&CORE measurement quantity $RH_{liq}$. The main challenge in this intercomparison is to ensure data comparability, given the relatively small number of CARIBIC ( 500 flights) compared to the large amount of MOZAIC&CORE data (together  60,000 flights). The relatively small number of CARIBIC data points could introduce non-negligible uncertainties in

the statistical comparison due to the natural variability of UT/LMS $H_2O$ caused by competing transport, chemical, and mixing processes near the tropopause. They affect particularly the UT and the extratropical transition layer (exTL) (Zahn et al., 2014), whereas $H_2O$ above the exTL exhibits much smaller variations (Gettelman et al., 2011). To minimize this uncertainty, a careful temporal and spatial sampling is performed using a geophysically-based coordinate system known to reduce sampling biases (Millan et al., 2023). The methodology is summarized in Figure 3.

Spatially, the $H_2O$ data are sampled relative to the TTP ($\Delta z_{\mathrm{TTP}}$) obtained from ERA5. The sampling starts 1 km below the TTP, with a vertical resolution of 0.25 km, and is done in equivalent latitude ranges ($\Delta\varphi_{\mathrm{EQ}}$) of 5°. Cases with double tropopauses were excluded from the analysis. We found that incorporating the dynamical tropopause, defined as 2 potential vorticity units (PVU; $1\,\mathrm{PVU} = 10^{-6}\,\mathrm{K}\cdot\mathrm{m}^2\cdot\mathrm{s}^{-1}\cdot\mathrm{kg}^{-1}$), neither improved nor worsened the quality of the resulting intercomparison. Instead of using latitudinal mean values of $H_2O$, means corresponding to ERA5 equivalent latitude are calculated,

because the potential vorticity based equivalent latitude characterizes the latitudinal air mass origin in the UTLS (Gettelman et al., 2011), which is correlated with corresponding $H_2O$ amounts to some extent.

On a temporal scale, we calculate monthly means ($\Delta\tau_{\mathrm{month}}$) to account for the seasonal variability of UT/LMS $H_2O$ (Zahn et al., 2014; Petzold et al., 2020). Since $H_2O$ data of MOZAIC&CORE are derived from $RH_{liq}$ measurements, which depend on temperature, we further divide each bin ($\Delta z_{\mathrm{TTP}}$, $\Delta\varphi_{\mathrm{EQ}}$, $\Delta\tau_{\mathrm{month}}$) into temperature ranges ($\Delta T$) of 1 K. This temperature

sampling ensures that any potential discrepancies in the sampled $RH_{liq}$ due to differences in the temperature distribution between MOZAIC&CORE and CARIBIC are accounted for. Additionally, the temperature at a certain height level can correlate with atmospheric conditions, which in turn can influence the $H_2O$ values. Although we also performed a potential temperature sampling, it did not yield significant differences in the final intercomparisons and was therefore not used in the final sampling strategy. The temperature sub-sampling at specific heights essentially achieves the same effect as the potential temperature

sampling, given the relatively stable flight pressure levels.

To enhance the statistical comparability, we incorporate ozone measurements that were conducted during the majority of





MOZAIC&CORE, and CARIBIC flights. Bethan et al. (1996) and Sprung and Zahn (2010) demonstrated that ozone measurements are effective in delineating the structure of the LMS. Above the tropopause, the characteristic increase of ozone concentrations makes it suitable as stratospheric tracer and can be indicative for stratospheric air masses to resolve UT-LMS

mixing processes. Therefore, for each bin sampled above the thermal tropopause, we further sub-sample the air masses based on ozone concentrations in steps of $\Delta O_3$ = 10 ppbv, spanning the range from 60 to 700 ppbv.

The binning process ($\Delta z_{TTP}$, $\Delta \varphi_{EQ}$, $\Delta \tau_{month}$, $\Delta T$, $\Delta O_3$) is conducted for each flight individually (Figure 3a; Figure 3b - Step 1). For every measurement that falls into a respective sampling bin and flight, we compute the sampling means $\overline{RH_{liq}^{fl}}$ (see Figure 3a). To derive a relevant sampling mean, a minimum of 10 data points is required per bin and flight. Subsequently, for

each of these bins, the arithmetic mean is computed from all the mean values obtained from the individual flights:

$$\overline{RH_{liq}^*} = \frac{1}{N_{fl}^*} \sum_{i=0}^{N_{fl}} \overline{RH_{liq}^{fl}[i]}, \tag{1}$$

where $N_{fl}^*$ is the total number of flights for a certain sampling bin.

Data from different flights are weighted equally to mitigate the potential influence of measurements from individual flights that might have a larger number of observations compared to most other flights. This equal weighting is particularly crucial for

the sampling of CARIBIC data, where some bins may contain data from 10 flights or less, making it necessary to ensure fair representation of all flights in the analysis.

In the final step, to ensure a sufficient number of bins with enough data to serve as a valid reference, we derive weighted seasonal averages (DJF, MAM, JJA, SON) from the monthly means, and 10° weighted $\Delta \varphi_{EQ}$ means from the 5° $\Delta \varphi_{EQ}$ means. Additionally, we calculate the weighted mean across all 1 K $\Delta T$ and 10 ppbv $\Delta O_3$ ranges. In this averaging process,

each bin is weighted based on the number of CARIBIC flights ($N_{fl}^{CA^*}$) that contribute to the mean values (see Eq. 1) in every 1 K $\Delta T$, 10 ppbv $\Delta O_3$, 5° $\Delta \varphi_{EQ}$, and monthly range:

$$\overline{RH_{liq}} = \frac{1}{N_{fl}^{CA}} \sum_{i=0}^{n} \overline{RH_{liq}^*}[i] \times N_{fl}^{CA}[i], \quad \text{with} \quad N_{fl}^{CA} = \sum_{i=0}^{n} N_{fl}^{CA^*}[i], \tag{2}$$

where n is the number of bins that contribute to $\overline{RH_{liq}}$.

This weighted averaging approach ensures that each bin is appropriately represented in the final intercomparison, taking into

account the relatively small number of flights available in the CARIBIC data set compared to MOZAIC&CORE. For instance, in the calculation of seasonal means from the respective monthly means, a higher number of flights during a particular month by CARIBIC compared to MOZAIC&CORE could lead to an over-representation of that month in the CARIBIC seasonal mean, affecting the overall intercomparison results. Since $H_2O$ in the extratropical UT/LMS exhibits non-negligible variations on a monthly scale (Zahn et al., 2014; Kunz et al., 2008), this weighting is crucial to ensure a fair comparison between the

datasets and to obtain reliable and meaningful results. Overall, this approach was found to significantly improve the accuracy of the statistical comparability of the datasets.

By applying the mapping approach described above, sampling bins that contain too few flights are excluded, as these could lead to higher uncertainties due to the natural variability of $H_2O$ in the UT/LMS despite the use of the geophysically-based





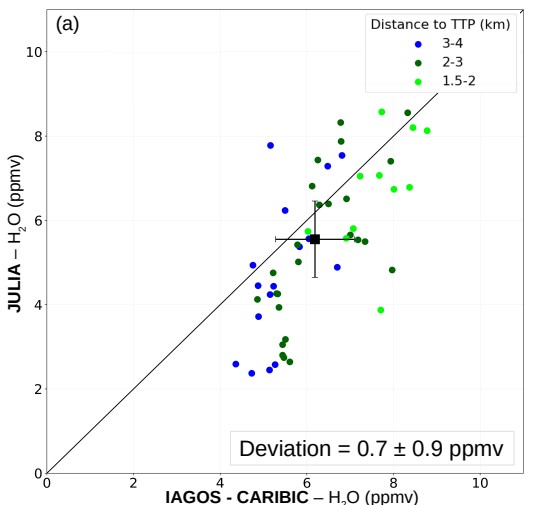
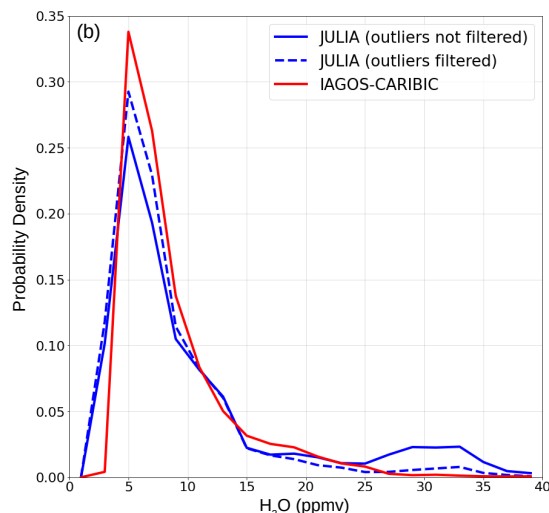

**Figure 4. Statistical intercomparison of IAGOS-CARIBIC with JULIA H$_2$O.** (a) LMS H$_2$O mixing ratios of IAGOS-CARIBIC versus JULIA, calculated based on the methodology described in Section 3.1 and 3.2; the color code denotes the distance to the TTP (see legend). The mean bias and the corresponding standard deviation based on all sampling bins is indicated by the black dot and the bar, respectively. (b) Based on single measurements from all sampling bins in (a), the frequency distribution is shown both with and without the filtering of strong outliers.

coordinate system. This uncertainty is expected to decrease with height above the tropopause, as has been shown for H$_2$O based on the calculation of a trade-off factor between including more measurements versus adding more geophysical variability (due to year-to-year or longitudinal variations). This trade-off factor shows that much less measurements are needed to constrain the mean H$_2$O value in a certain bin the higher above the tropopause it lies (Hegglin et al., 2008), providing confidence to our approach taken. Thus, the minimum number of flights required for a sampling bin to be included in the final intercomparison varies with the height relative to the TTP. Specifically, the minimum number of flights linearly decreases from 15 in the lowest

vertical range below the TTP (-1 to -0.75 km) to 4 flights at 3 km and higher above the TTP.

Despite the relatively limited number of flights available in the CARIBIC data set, a considerable number of sampling bins ($\Delta z_{TTP}$ = 0.25 km, $\Delta \varphi_{EQ}$ = 10°, $\Delta \tau_{season}$) contain a sufficient number of data and flights through the years to represent the multi-annual climatological state and thus enable a reliable intercomparison with MOZAIC&CORE (as detailed in Section 3.3).

**3.2 Intercomparison of IAGOS-CARIBIC and JULIA H$_2$O**

H$_2$O data from CARIBIC serve as reference for evaluating the extensive dataset of MOZAIC&CORE. To ensure the high quality of the CARIBIC data, the performance of the CARIBIC H$_2$O measurements (combination of WaSul and FPH; see Section 2.1.2) is validated by comparison with high precision instruments, such as FISH, compiled in the research aircraft data set JULIA (Section 2.1.3). In the past, joint flights of the WaSul sensor and the FISH hygrometer were conducted, enabling





a direct comparison of $H_2O$ measurements from both instruments (Meyer et al., 2015; Tatrai et al., 2015; Rolf et al., 2023). While notable differences outside the expected noise level were observed between the two instruments in dry stratospheric air masses of 10 ppmv and less, no systematic bias in either the dry or wet direction was identified (Tatrai et al., 2015). However, because of the additional operation of an FPH, further investigation is needed to assess the agreement between CARIBIC and JULIA $H_2O$ in the LMS.

### 3.2.1  Preparation of the datasets

For the comparison of CARIBIC and JULIA $H_2O$ data in the LMS, the binning strategy described in Section 3.1 and depicted in Figure 3 is employed. In the comparison, only sampling bins at distances of at least 1.5 km above the TTP are taken. This reduces the impact of the natural $H_2O$ variability in the comparison, which is higher close to the TTP and in the UT, thereby improving the accuracy and reliability of the results.

The higher sampling altitude of JULIA results in the sampling of different air masses, which might correspond to different $H_2O$ concentrations, compared to CARIBIC. Therefore, we only consider CARIBIC and JULIA measurements between 10.5 and 12.5 km and a mean height difference below 0.5 km between the respective sampling bins to ensure consistency. Moreover, in our data sampling process, we use potential temperature instead of temperature, further reducing the effect of the natural $H_2O$ variability.

The JULIA data are subjected to a filtering criterion that excludes cases with $RH_{ice}$ values larger than 80 % (see Section 2.1.3). This criterion is applied because the JULIA data include total water (=gas phase + cloud ice particles). Using values below 80 % $RH_{ice}$ ensures that in-cloud measurements are excluded. For consistency in the comparison, we also exclude $RH_{ice}$ values above 80 % in the CARIBIC data set.

Last, but not least, to reduce the influence of strong outliers, CARIBIC and JULIA sampling bins are excluded from the mean
$H_2O$ values derived according to Equation 1, if the difference between the sampling bins is 10 ppmv or higher. Such deviations are beyond the expected error range in the LMS (Tatrai et al., 2015) and likely stem from varying atmospheric conditions during the respective measurements, resulting in notable differences in $H_2O$.

### 3.2.2  Intercomparison

The final statistical comparison of the CARIBIC and JULIA $H_2O$ datasets with all criteria described in Section 3.2.1 applied, is
shown in Figure 4 (a), where each point represents one sampling bin. The mean values and corresponding standard deviations of all bins are indicated by the square dot and the error bar, respectively. The comparison shows a scattering of the sampled mean $H_2O$ values along an ideal regression line, but, on average, a good agreement, with a mean difference of CARIBIC compared to JULIA of $(0.7 \pm 0.9)$ ppmv. The scattering might be attributed to the limited amount of data from both products and the potential over-representation of anomalous atmospheric conditions in the JULIA $H_2O$ data, despite the efforts to filter
out strong outliers. Figure 4 (b) displays the probability density function (PDF) based on all bins given in (a). Without the filtering, a significant amount of anomalously high $H_2O$ is present in the JULIA data base, which are mostly filtered out with this approach (blue dashed line) while for CARIBIC, this filtering approach does not cause significant differences (thus, only





**Figure 5. Statistical intercomparison of H$_2$O from IAGOS-MOZAIC&CORE with IAGOS-CARIBIC** . Sampling bin means of $\overline{\overline{RH_{liq}}}$ (left column), $\overline{RH_{ice}}$ (middle column) and $\overline{H_2O}$ (right column) for IAGOS-MOZAIC (top) and IAGOS-CORE (bottom) versus IAGOS-CARIBIC. The sampling bins are derived following the methodology described in Section 3.1; the color code denotes the distance to the tropopause, the symbols the corresponding season.

filtered data displayed in Figure 4b).

Systematic differences can be found for sampling bins where JULIA indicates H$_2$O of less than 6 ppmv. Bins with JULIA
indicating H$_2$O of less than 3 ppmv occurred during the winter season in polar air masses with very low cold point tropopause
temperatures. It is not unlikely that these are cases of anomalously low H$_2$O that were purposefully measured during certain
campaign flights and meteorological conditions, and are thus over-represented in JULIA. However, it cannot be argued whether
this small mean deviation is a result of different atmospheric sampling strategies between campaign and commercial aircraft
flights, despite the strict filter conditions, or a small systematic bias. Nevertheless, the discrepancies shown in Figure 4b are
within the known uncertainties of the H$_2$O instruments used in CARIBIC and JULIA flights (Table 1) and thus the CARIBIC
H$_2$O data are suitable for the comparison with MOZAIC&CORE, as described in the next section.



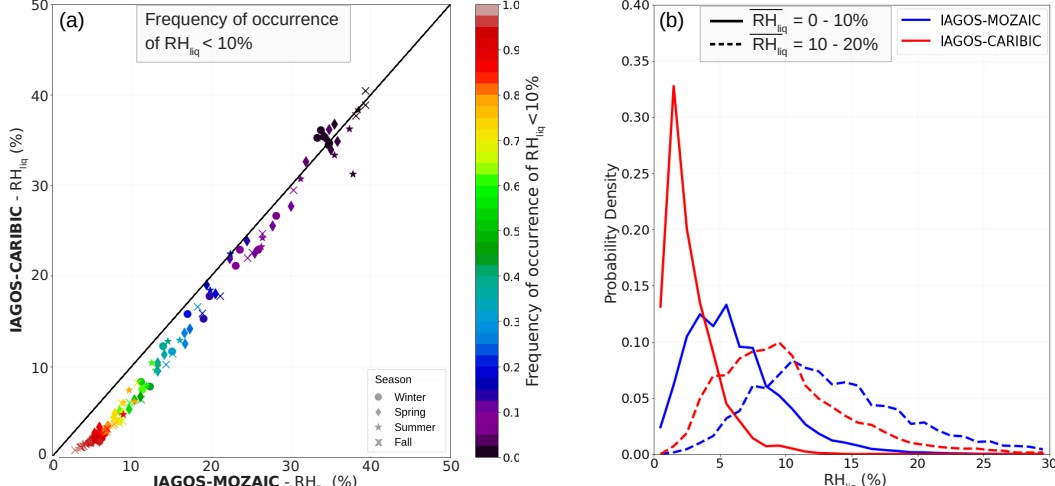

**Figure 6.** (a) For IAGOS-MOZAIC, the plots show the same sampling bins as in Figure 5. The color code indicates the frequency of occurrence of single measurements below a threshold of 10% by IAGOS-MOZAIC that contribute to the shown mean values. (b) For all bins in Figure 5 with $\overline{RH_{liq}}$ of 0-10% and 10-20%, the corresponding occurrence frequencies of $RH_{liq}$ measurements is shown for IAGOS-MOZAIC and -CARIBIC.

### 3.3 Evaluation of IAGOS-MOZAIC&CORE H$_2$O

Figure 5 displays the intercomparison of the sampled mean values $\overline{RH_{liq}}$, $\overline{RH_{ice}}$, and $\overline{H_2O}$ (2) with respect to the season and the distance to the tropopause. Additionally, Figure A1 shows the same plots, but the data being sampled into 5 K means instead
of being averaged over all temperature ranges. MOZAIC and CORE are regarded separately to examine the agreement between them. A majority of the measurements contributing to the sampled mean values originate from the extratropics between 30 and 80°N for which this intercomparison is valid for.

In the comparison of MOZAIC to CARIBIC data, a distinct relationship is observed. When examining bins below the TTP where $\overline{RH_{liq}}$ values are mostly above 30 %, MOZAIC and CARIBIC show a good agreement. This agreement is attributed to
the infrequent occurrence of $RH_{liq}$ values of 10 % and less, which tend to be biased (Neis et al., 2015a), and therefore do not significantly impact the mean values in this height range.

Figure 6a illustrates the correlation between the relative number of MOZAIC measurements below 10 % $RH_{liq}$ and the bias between MOZAIC and CARIBIC. For MOZAIC and CARIBIC, Figure 6b displays the PDFs from all measurements that fall within the 0 - 10 % and 10 - 20 % $\overline{RH_{liq}}$ ranges of MOZAIC in Figure 6a. Notably, the bias is prominent when the amount
of measurements below 10 % $RH_{liq}$ is higher than 20 % (Figure 6a), while bins with fewer measurements below 10 % $RH_{liq}$ show little to no significant biases. This correlation pattern is also observed for CORE (not shown).

The sampling bins with $\overline{RH_{liq}}$ values below 20 % exhibit significant systematic moist biases, with relative differences of 100 % or more for $\overline{RH_{liq}}$ of 10 % and less. For layers closer to the TTP, there is a lower but still noteworthy bias. During



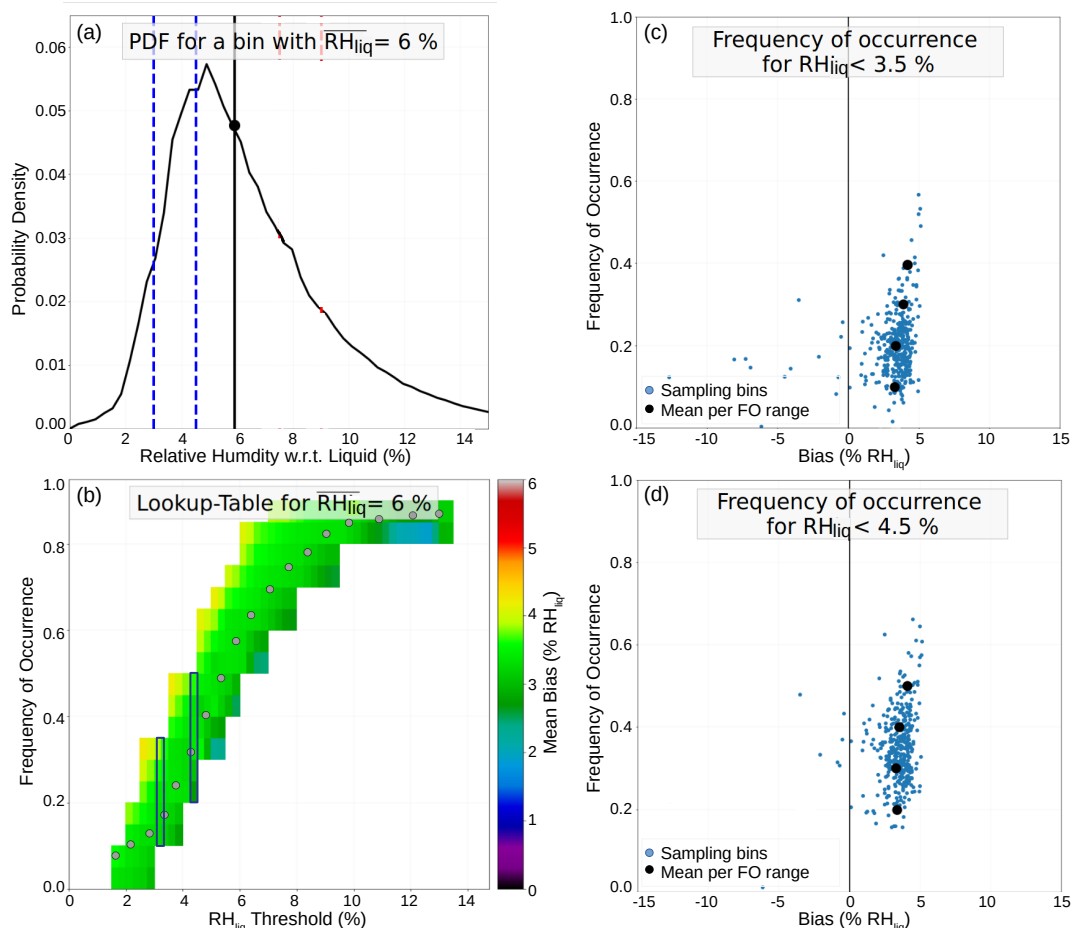

**Figure 7. Application example of the IAGOS adjustment algorithm**. (a) Example $RH_{liq}$ frequency distribution of a sampling bin with a mean of $\overline{RH_{liq}}$ = 6 %. For this mean value, (b) shows the lookup-table with the values used as adjustment based on different $RH_{liq}$ thresholds and the corresponding frequency of occurrences (FOs). For the two thresholds indicated by the blue dotted lines in (a), the plots (c) and (d) exemplary show the derivation of the mean bias (black dots) based on the weighted mean of the sampled mean values (blue dots).

summer, certain bins have $\overline{RH_{liq}}$ values of 15 % or less already in the first kilometer above the TTP, whereas the same levels
relative to the TTP during other seasons indicate much higher values. This seasonality is attributed to the higher temperatures observed in the respective sampling bins during the summer season. Conversely, in the winter season, a considerable number of MOZAIC&CORE measurements in the lowest kilometer above the TTP are higher than 10 % (as shown in Figure 6a), thereby providing more reliable mean values. Consequently, biases of the mean values are in the range of only about 1 to 3 % $RH_{liq}$ during this season close to the TTP.

For the comparison of CORE and CARIBIC, we used data between 2018 and 2022 only. This specific time frame was chosen because before 2018, a systematic failure of the temperature sensor at the humidity capacity sensor occurred. Although the



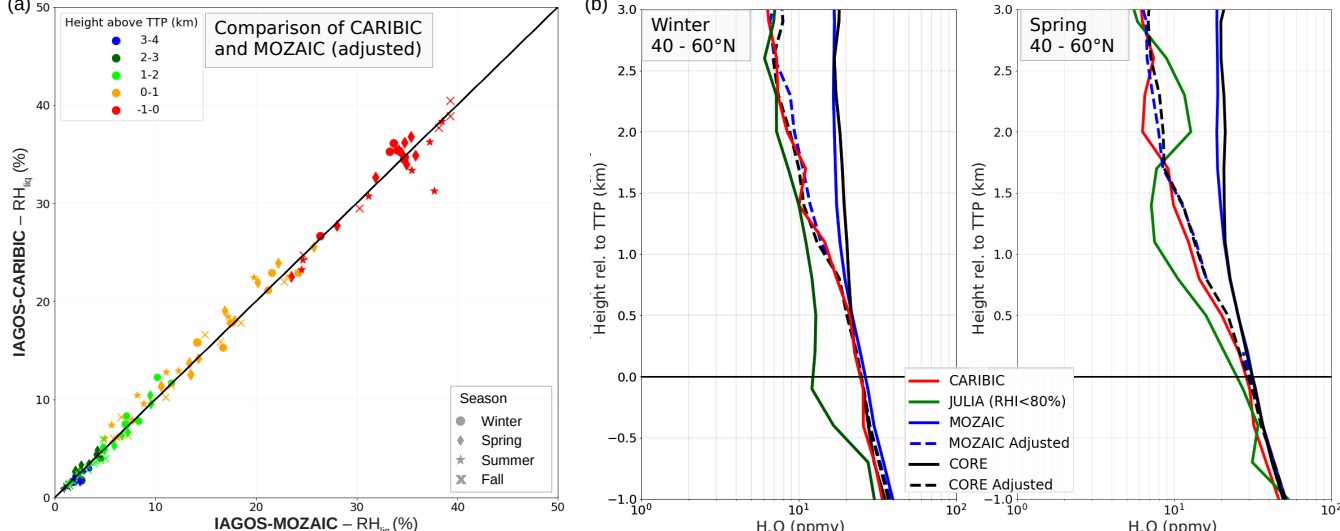

**Figure 8. Application of the adjustment algorithm on the IAGOS data**. Panel (a) shows a comparison of the same sampling bins between IAGOS-MOZAIC and IAGOS-CARIBIC as shown in Figure 8 but with the adjustment algorithm applied to the mean values. Panel (b) shows two mean vertical UT/LMS $H_2O$ profiles of IAGOS-CARIBIC, JULIA, and IAGOS-MOZAIC&CORE (adjusted and unadjusted).

resulting error was adjusted afterwards, we decided to utilize only data with the highest quality unaffected by this issue, to ensure that no systematic biases affect the analysis.

On average, the comparison of CORE and CARIBIC shows similar mean biases in the LMS, similar to what the comparison
with MOZAIC revealed. However, for CORE, there is a stronger variation of the bins compared to MOZAIC, which can be attributed to differences in the temporal coverage (2018-2022 and 1995-2022 for CORE and CARIBIC, respectively, due to the large year-to-year variability of UT/LMS $H_2O$ (Kunz et al., 2008).

# 4    Adjustment of IAGOS-MOZAIC&CORE LMS $H_2O$ to IAGOS-CARIBIC

## 4.1    Adjustment methodology

Based on the mapping approach used to make the different $H_2O$ data sets comparable (Section 3.1), the next step is to apply an adjustment algorithm to the biased MOZAIC&CORE $H_2O$ data, using CARIBIC $H_2O$ as reference. For individual measurements, a fixed bias as a function of $RH_{liq}$ cannot be determined. This is due to the sensor offset drift at 0% $RH_{liq}$ that occurs between the ICH calibrations, which are conducted every three months (Petzold et al., 2020). An in-flight calibration is used to adjust this sensor shift (Smit et al., 2008), but uncertainties in the in-flight calibration (as discussed in Smit et al. (2008)) may
cause a nonlinear behavior in the bias of single measurements as a function of $RH_{liq}$ between the calibrations. Consequently, it is also not feasible to directly use data from campaign flights with ICH and high-precision $H_2O$ instruments like FISH to adjust ICH data measured onboard IAGOS flights.





Because of the reasons mentioned above, the primary objective is to focus on adjusting sampling bin mean values $\overline{\mathrm{RH_{liq}}}$ (Section 3.3), which consist of a large number of measurements in the order of several thousand that represent a climatological

state,. One approach would be to fit the PDFs of MOZAIC&CORE to the ones of the reference dataset, CARIBIC. However, because of a relatively small number of CARIBIC data, the corresponding PDFs of the CARIBIC sampling bin mean values (i.e. the ones shown in Figure 5) are often not smooth, in contrast to the smoother distributions observed in the MOZAIC&CORE dataset. This disparity poses a challenge when attempting to fit the PDFs straightforwardly. To address this issue, an alternative approach is adopted, which is described below.

For every sampling bin according to Equation 1, each of the corresponding $RH_{liq}$ frequency distributions is sliced into segments. For every of these segments, the frequency of occurrence (FO) of measurements falling below specific thresholds is derived, i.e. $\mathrm{FO} = \sum_{ii=0}^{RH_{liq}^{thres.}} \mathrm{PDF}[ii]$, as shown for the 10 % threshold in Figure 6a. This principle is illustrated in Figure 7a for one specific $RH_{liq}$ PDF with a sampling mean value of $\overline{\mathrm{RH_{liq}}} = 6\%$. For all sampling bins that fall within the range 5.5 to 6.5 % $RH_{liq}$, Figure 7c and d illustrate the FO of two specific thresholds ($\sum \mathrm{FO}$ for $RH_{liq} < 3\%$ and $< 4.5\%$) and the respective

biases of MOZAIC compared to CARIBIC. Again, MOZAIC and CORE were treated separately but the results also account for CORE.

The corresponding mean biases as a function of the FO (dashed regression lines in Figure 7c,d) exhibit a robust relation with the FO, and the variations along the bias regression line are within $\pm 0.5$ % $RH_{liq}$. For $\overline{\mathrm{RH_{liq}}} = 6\%$, a broader distribution, i.e. a higher FO for the two thresholds shown in Figure 7c and d, correspond to a higher bias. Similar plots like the ones in Figure 7

, but for a mean value of $\overline{\mathrm{RH_{liq}}} = 15$ %, are presented in Figure A2.

By performing this approach for certain $\overline{\mathrm{RH_{liq}}}$ ranges, like it was done for the 5.5 – 6.5 % range, the mean biases as a function of the respective FOs are calculated. Furthermore, a moving average step size of 0.25 % $RH_{liq}$ is applied for $\overline{\mathrm{RH_{liq}}}$, e.g. 5.5 - 6.5 % $\overline{\mathrm{RH_{liq}}}$, 5.75 – 6.75 % $\overline{\mathrm{RH_{liq}}}$, and so on.

Using this approach, we obtain the mean biases $\Delta RH_{liq}$, as a function of (1) ranges of $\overline{\mathrm{RH_{liq}}}$ values, and (2) $RH_{liq}$ thresholds

and the corresponding FOs. Subsequently, we apply interpolation to the mean deviations between various ranges of average $\overline{\mathrm{RH_{liq}}}$ values to mitigate fluctuations stemming from uncertainties in the $RH_{liq}$ distribution. This approach yields a set of comprehensive lookup-tables containing the mean deviations for specific average $\overline{\mathrm{RH_{liq}}}$ values and the respective FO thresholds. An illustrative example of such a lookup-table for $\overline{\mathrm{RH_{liq}}} = 6\%$ is presented in Figure 7b. The two black boxes correspond to the biases of the two thresholds shown in 7c and d.

In the final adjustment, the mean bias is derived from the arithmetic mean of the biases indicated by different FO thresholds:

$$\overline{\Delta \mathrm{RH_{liq}}} = \frac{1}{n} \sum_{i=0}^{n} \Delta \mathrm{RH_{liq}}[i], \tag{3}$$

with n number of FO thresholds.

As an example, for the distribution shown in Figure 7a, the biases for different $RH_{liq}$ FO thresholds are represented by the black line in Figure 7b. To obtain the final mean bias $\overline{\Delta \mathrm{RH_{liq}}}$, the arithmetic mean of all these individual biases is calculate.

This approach ensures that we consider and incorporate the biases associated with various FO thresholds, leading to a more comprehensive and accurate assessment of the mean bias for the specific mean value $\overline{\mathrm{RH_{liq}}}$.



Applying the adjustment algorithm described in this section to the entire MOZAIC data set, i.e. using the sampled mean values in Figure 5, a good agreement to the CARIBIC data can now be found, as can be seen in Figure 8a, where the sampled $\overline{\mathrm{RH}_{\mathrm{liq}}}$ are closely distributed around the 1-to-1 line without obvious bias. Furthermore, Figure 8b illustrates two examples of vertically
resolved mean $H_2O$ from MOZAIC (blue) & CORE (black) (both not adjusted, solid lines, and adjusted, dashed lines) as well as CARIBIC (red) and JULIA (green) during the winter and spring seasons in the 40 - 60°N latitudinal region. At heights of 1km and more above the TTP (depending on the season), where MOZAIC&CORE exhibit significant biases, the adjusted mean values now reveal a good agreement with JULIA and CARIBIC within the expected variation range. This outcome signifies that the adjustment methodology provides reliable mean values independent of the season. For JULIA, lower values compared
to CARIBIC, MOZAIC, and CORE in the lowest 1 km above the TTP can be explained by the filtering of mean $H_2O$ values that correspond to ice relative humidity higher than 80 %to avoid sampling bins containing ice particles (as described in Section 2.1.3).

## 4.2    Application and uncertainties of adjusted IAGOS-MOZAIC&CORE H₂O

The adjustment of the MOZAIC&CORE-based $H_2O$ climatologies offers the advantage of resolving spatial and seasonal
variability at a higher resolution than would be possible with just the in-situ data sets of CARIBIC and JULIA. The adjustment of mean values requires a sufficient number of measurements, in order to provide a smooth PDF based on which the adjustment is performed. In the lower stratosphere (LS), variability in $H_2O$ increases with altitude towards the tropopause, necessitating a larger number of measurements to ensure the PDF is not skewed by outliers.

. To determine the necessary number of measurements, a Kolmogorov-Smirnov test (Berger and Zhou, 2014) is performed.
This test assesses whether the data fits a specific distribution, typically a Weibull distribution for the IAGOS $\mathrm{RH}_{\mathrm{liq}}$ data, with at least 95 % confidence. For the sampling strategy outlined in this study (Section 3.1), the required number of data points ranges from approximately 300 (LMS; $\Delta z > 2$ km) to 1000 (UT).

Despite the good agreement as shown in Figure 8, one has to keep in mind that for the resulting adjusted MOZAIC&CORE $\overline{\mathrm{H_2O}}$, uncertainties due to (1) measurement uncertainties and (2) uncertainties in the adjustment itself incorporate. For the
CARIBIC measurement uncertainty, in the derivation of $\mathrm{RH}_{liq}$, uncertainties of $H_2O$ (4 %), temperature (0.7 K; Benjamin et al., 1999) and pressure (1 hPa; Tang et al., 2005), are taken into account. The resulting relative bias in the error propagation is in the order of 7 %.

The uncertainty due to limitations in the method results from the small number of CARIBIC data and differences in the temporal coverage between the products, with $H_2O$ showing a large year-to-year variability in the LMS (Kunz et al., 2008).
Furthermore, differences in the geographical coverage might also induce uncertainties, despite that the sampling strategy should strongly decrease uncertainties in the comparison due to this reason.

The method uncertainty is determined like the following: The bias derivation (see Figure 7) is also performed for each season separately. In the next step, the standard deviation for each bias as a function of $\overline{\mathrm{RH}_{\mathrm{liq}}}$ and FO (see last section) is derived from the four seasonal means, and from these deviation, the mean standard deviation as a function of just $\overline{\mathrm{RH}_{\mathrm{liq}}}$. The mean
bias (averaged over all FO) as a function of unadjusted $\overline{\mathrm{RH}_{\mathrm{liq}}}$ is shown in Figure 9 (dashed line), with the red area indicating



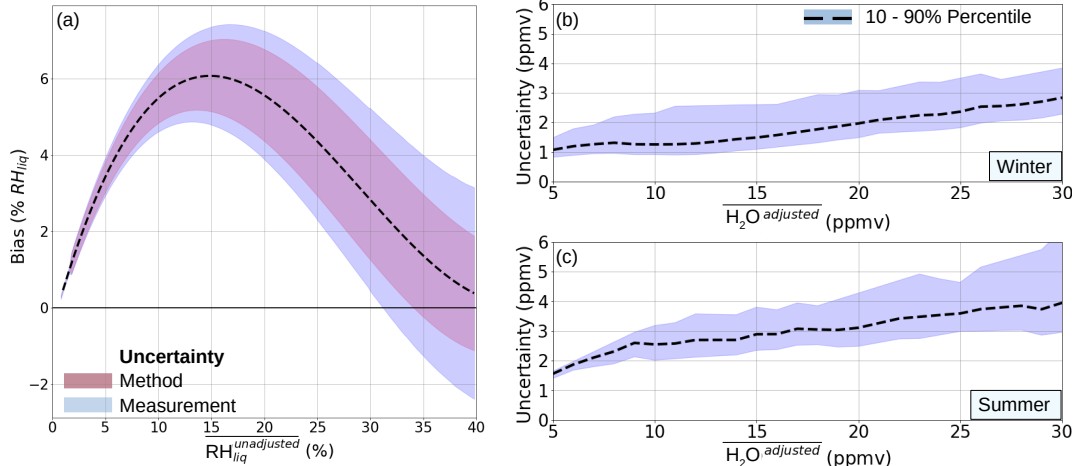

**Figure 9. Error Budget** (a) Mean bias derived from the mean of all FO thresholds as a function of $\overline{\mathrm{RH_{liq}}}$ (solid line) and the uncertainty estimate due to the adjustment method (red shaded) and due to measurement uncertainties (blue shaded). (b and c) For $\overline{\mathrm{H_2O}}$ derived from adjusted $\overline{\mathrm{RH_{liq}}}$, the mean bias (dashed line) and the 10 – 90 % percentile (blue shaded) is shown for winter (b) and summer (c).

the uncertainty due to the adjustment method and the blue area the additional uncertainty when measurement uncertainties are also taken into account. The uncertainty of the derived adjusted $\overline{\mathrm{H_2O}}$ vary depending on the corresponding $\overline{\mathrm{RH_{liq}}}$. During summer, $\overline{\mathrm{RH_{liq}}}$ in the LMS tend to be lower due to higher temperatures compared to winter, with lower $\overline{\mathrm{RH_{liq}}}$ having higher relative uncertainties. Based on all $\overline{\mathrm{RH_{liq}}}$ in the extratropical LMS, Figure 9b and c show the mean uncertainty and the 10 –

90 % percentile (dashed line and shaded area, respectively). For 10 ppmv for example, the mean bias (bias range) is 1.0 (0.8 – 2.3) ppmv and 2.4 (2.0 – 3.1) ppmv for winter and summer, respectively.

## 4.3    Adjusted UT/LMS H$_2$O climatologies

Multi-annual monthly means of adjusted H$_2$O, based on all MOZAIC&CORE data, are shown in Figure 10 for two $\Theta$ levels, 335 K (a-d) and 350 K (e-h). The magenta solid line indicates the mean 2 PVU line.

Over parts of N. America as well as SE. Asia, the monsoon related H$_2$O increase during the Northern hemisphere summer season (Nützel et al., 2019) is evident at $\Theta = 350$ K (Figure 10g), i.e. at a $\Theta$ level that corresponds to subtropical and tropical air masses at passenger aircraft altitude over these regions. Here, mean values are by a factor of 3 (N. America) to 10 (SE. Asian) higher, compared to other regions at the same latitudes.

From fall to spring, highest values in the mid-latitudes at 335 K can be found over the N. Atlantic. Higher H$_2$O amounts over

the Atlantic during the winter half of the year, higher compared to the continental regions, correlate with the high low pressure activity over this area, and resulting large scale uplift of moist and relatively warm air masses (UT) and potential isentropic mixing of moisture into the LS. Enhanced isentropic mixing into the LS over the Atlantic was found to occur in relation to warm conveyor belt outflow (Kunkel et al., 2019), based on measurements during the WISE (Wave-driven ISentropic Ex-





**Figure 10.** $H_2O$ mixing ratio climatologies. Monthly means (January, April, July and October) for two $\Theta$ level. In magenta, solid, dashed, dotted, and dash-dotted, the mean position of the 2, 4, 6 and 8 PVU line is indicated.

change) campaign.

For the N. Atlantic (50 - 70°N and 5 - 65°W), Figure 11a-d show adjusted $\overline{H_2O}$ climatologies, given in coordinates of equivalent latitude and potential temperature difference relative to the TTP ($\Delta\Theta$). Close to the TTP ($\pm$ 20 K), a strong annual $H_2O$ cycle can be observed. Along the TTP, the $H_2O$ varies between 20 to 30 ppmv (winter; $\Theta$ = 315K – 325), and 100 ppmv (summer; $\Theta$ = 325 – 340 K).

Investigating the annual cycle along isentropic levels in the LMS, at 340 K, a distinct increase during the summer half of the year can be found. During January, $H_2O$ is in the range of 5 – 10 ppmv and increases to 15 - 70 ppmv during July, with a strong gradient along $\Delta\varphi_{EQ}$ ranges, indicating the potential isentropic transport of $H_2O$ from the subtropical regions into the mid-latitudinal LMS. At 350 K, the annual cycle is less pronounced compared to the levels below, with a variation from mostly




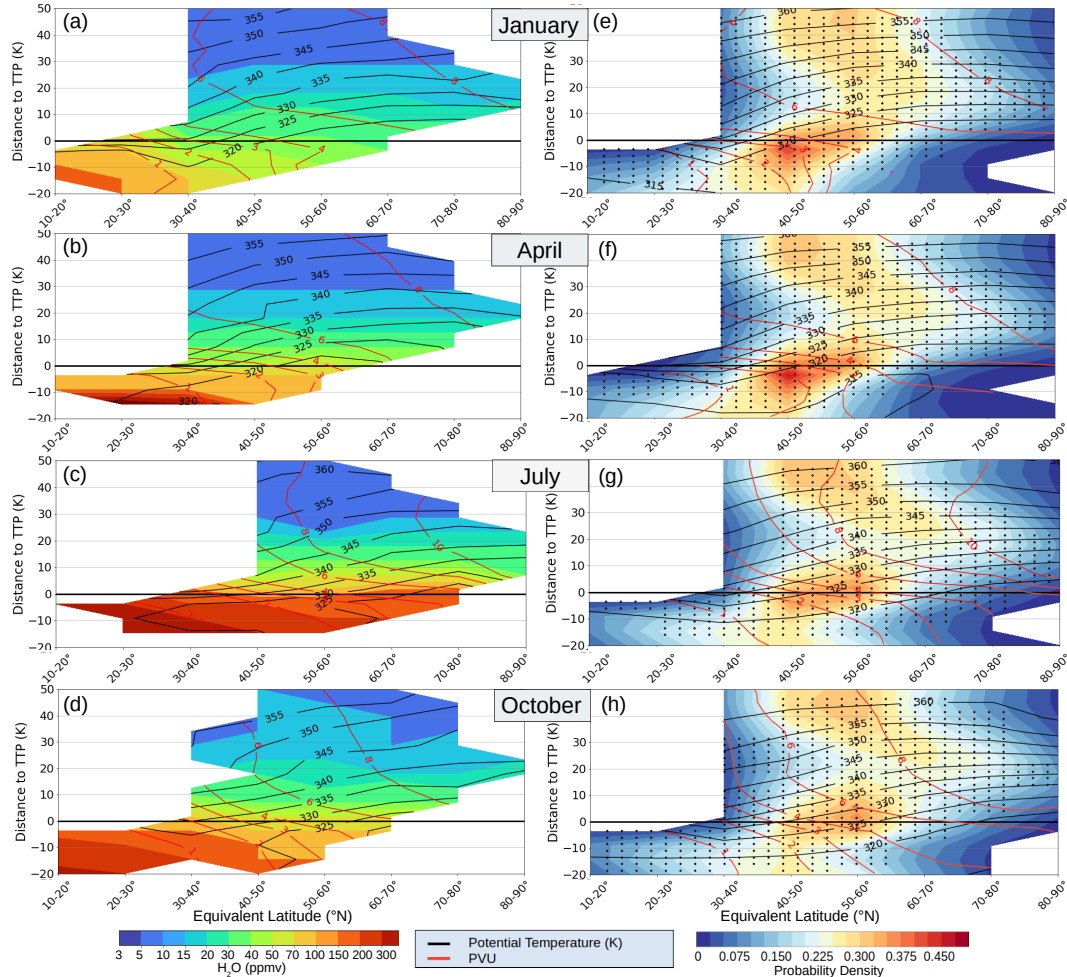

**Figure 11. Adjusted UT/LMS H$_2$O mixing ratio climatology for the North Atlantic region (50-70°N and 5-65°W).** (a-d) show the multi-annual monthly means of the adjusted H$_2$O . (e-h) Occurrence frequency of sampling bins of equivalent latitude and potential temperature difference ($\Delta\Theta$) to the TTP (the resolution is 5 K and normalized per $\Delta\Theta$ range). The black dots indicate sampling bins for which MOZAIC&CORE provide data from at least 20 flights.

5 - 7 ppmv along all $\Delta\varphi_{EQ}$ during winter, to an increase to 15 - 20 ppmv during summer in the southernmost $\Delta\varphi_{EQ}$ range covered (40 - 50°N).

At levels of $\Delta\Theta$ of around 20 K above the TTP, the highest values can be found during fall, with a maximum during October (Figure 11d), in contrast to a slight decrease from summer to fall in the LMS close to the TTP ($\Delta\Theta$ <10 K) and in the UT ((Figure 11c,d). An increase of the exTL height during the fall season is a well-known feature for the northern extra-tropics, as reported in various studies (e.g. Zahn et al., 2014).

The reasons for the seasonal variability described in this section are not aimed to be discussed in detail in this paper (how-



ever, see e.g. Gettelman et al., 2011; Zahn et al., 2014) but will be the focus of future publications based on the adjusted MOZAIC&CORE climatologies.

Last but not least, we want to examine how well the climatology shown in Figure 11a-d cover the UT/LMS over the North Atlantic, given that passenger aircraft fly on constant altitudes and might avoid certain weather conditions. In order to investigate this, a sampling of $\Delta\Theta$ and $\Delta\varphi_{EQ}$ is applied to the ERA5 data. The probability density (normalized per $\Delta\Theta$) based on ERA5

data from all vertical levels is shown in Figure 11. The dotted areas in the plots illustrate the sampling bins where IAGOS provides data from at least 20 flights. Overall, a good coverage can be observed. During winter and spring, only air masses in the UT below $\Delta\Theta$ = -15 K are not well covered (Figure 11a,b). During summer and fall (Figure 11d,e), air masses with an equivalent latitude of 30 - 40°N, i.e. of subtropical origin, are mostly not covered in the LMS which accounts to 20 – 30 % of all air masses. However, the most frequent air masses ($\Delta\varphi_{EQ}$ = 50 – 80°N) are covered by IAGOS also during summer season.

## 440  5  Conclusion and Outlook

This study presented an algorithm to adjust $H_2O$ climatologies in the lowermost stratosphere (LMS) measured with the compact IAGOS humidity capacity sensor (ICH) operating onboard MOZAIC&CORE to the sophisticated measurements from IAGOS-CARIBIC. First, a statistical comparison of MOZAIC&CORE with CARIBIC $H_2O$ was conducted, selecting CARIBIC as the reference dataset due to its advanced instrumentation and similar spatial and temporal distribution. Although CARIBIC has

a limited number of about 500 flights compared to around 60.000 flights by MOZAIC&CORE combined, it still provided a sufficient number of measurements for a valid intercomparison in the extratropical northern hemisphere.

For the comparison, a mapping approach was utilized, applying a sampling of measurements into bins of similar origin, considering equivalent latitude, season, and height relative to the tropopause, to derive corresponding mean $RH_{liq}$ values ($\overline{RH_{liq}}$). Initially, CARIBIC data were compared with high-precision campaign measurements summarized in the JULIA data base. It

was demonstrated that CARIBIC measurements can detect low stratospheric $H_2O$ concentrations of 10 ppmv or less, making them suitable for intercomparison with MOZAIC&CORE. However, it has to be regarded that JULIA measurements were typically conducted under specific atmospheric conditions, unlike the more statistically representative sampling by IAGOS flights. Consequently, JULIA data were excluded from the comparison and adjustment of MOZAIC&CORE, as including them might introduce greater uncertainties rather than providing additional benefits from having more data.

The comparison between MOZAIC&CORE with CARIBIC showed good agreements in the (extratropical) upper troposphere. However, above the tropopause, the average values were generally biased, with the discrepancy increasing with distance to the tropopause and reaching up to 300% relative differences for H2O at around 5 ppmv. This systematic bias in the lower stratosphere was attributed to limitations of the ICH sensor, which loses sensitivity below approximately 10% RHliq. Despite this, the sensors consistently performed well for mean values above 30 ppmv.

Subsequently, using the mapping approach, a method was developed to adjust $\overline{RH_{liq}}$ from MOZAIC&CORE to those from CARIBIC. The biases were quantified as a function of $\overline{RH_{liq}}$, enabling the adjustment of MOZAIC&CORE $H_2O$ climatologies with an uncertainty of approximately 1 ppmv (winter) to 2.5 ppmv (summer) for mean values of 10 ppmv and less.



A caveat is that the adjustment of $\overline{\mathrm{RH_{liq}}}$ is based on a statistical comparison of the small CARIBIC reference dataset with the much larger MOZAIC&CORE dataset. This introduces a small systematic error due to the limited representativeness of the
CARIBIC dataset. This representativeness error could be neglected for studies of variability and transport processes, but for $H_2O$ trend analyses this error must be considered and quantified. Nevertheless, due to the lack of in-situ measurements in the UT/LMS, the adjusted climatologies provide better resolution of temporal and spatial variability of UT/LMS $H_2O$ compared to other in-situ or space-borne datasets. This will contribute to a better understanding of the $H_2O$ variability in the extratropical UT/LMS and its connection to various transport and mixing processes. Based on the adjusted $H_2O$ climatologies, upcoming
studies will investigate the contribution of different transport mechanisms to the $H_2O$ variability, using backward trajectories and simulations of (de-)hydration of air masses along their pathways. This enhanced understanding of the $H_2O$ variability and its corresponding transport mechanisms is crucial for improving the quality of model simulations concerning current and future $H_2O$ concentrations in the UT/LMS and their impact on the radiative forcing in a warming climate.



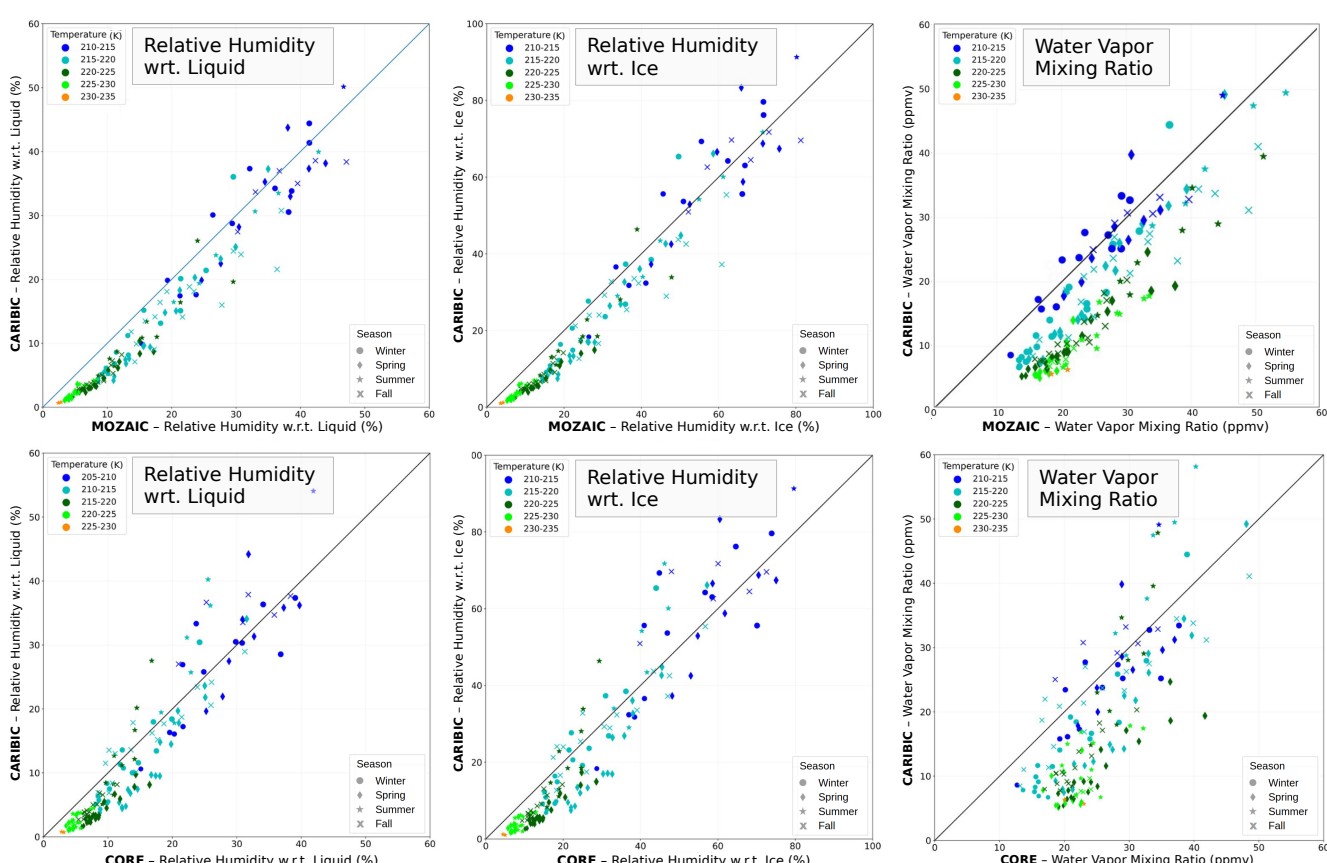

**Figure A1.** Intercomparison of sampled mean values of RH$_{liq}$, RH$_{ice}$ and H$_2$O for IAGOS-MOZAIC (top) and CORE (bottom) with IAGOS-CARIBIC. Instead of the height relative to the tropopause as shown in Figure 5, the colors indicate temperature ranges.



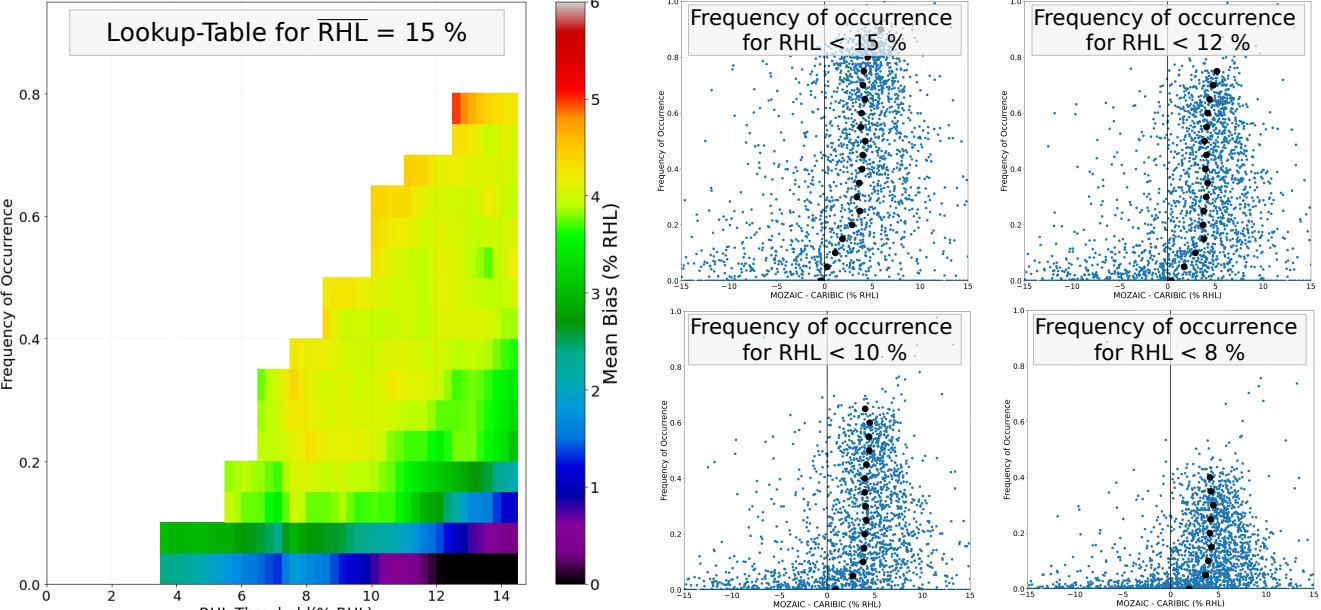

**Figure A2.** (a) Lookup-table for a mean value $\overline{RH_L}$ of 15 % which will be used to adjust potentially biased mean values of 15 %. (b) Corresponding frequency of occurrence for exemplary thresholds that are the basis for the final lookup-table

*Author contributions.* PK developed the methodology, performed the analysis, and wrote the manuscript. CR, MK, HB and AZ contributed
to the development of the mapping and adjustment approach. HB and AZ provided and helped with analysing the IAGOS-CARIBIC dataset.
AP, SR and YL provided and helped with analysing the IAGOS-CORE dataset

*Competing interests.* At least one of the (co-)authors is a member of the editorial board of Atmospheric Chemistry and Physics. The authors
have no other competing interests to declare.

*Acknowledgements.* The study was funded by the Deutsche Forschungsgemeinschaft (DFG, German Research Foundation) – TRR 301
– Project-ID 428312742. Parts of the study were also funded by ESA (contract no. 4000123554) via the Water Vapour Climate Change
Initiative (WV_cci) project phase 2 of ESA's Climate Change Initiative (CCI). We acknowledge the European Centre for Medium-Range
Weather Forecasts (ECMWF) for their ERA-5 meteorological data. MOZAIC/CARIBIC/IAGOS data were created with support from the
European Commission, national agencies in Germany (BMBF), France (MESR), and the UK (NERC), and the IAGOS member institutions
(http://www.iagos.org/partners). The participating airlines (Lufthansa, Air France, Austrian, China Airlines, Hawaiian Airlines, Air Canada,
Iberia, Eurowings Discover, Cathay Pacific, Air Namibia, Sabena) supported IAGOS by carrying the measurement equipment free of charge
since 1994. The data are available at http://www.iagos.fr thanks to additional support from AERIS.



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
