# Peer review of "Technical Note: Water Vapor climatologies in the extra-tropical Upper Troposphere and Lower Stratosphere derived from a Synthesis of Passenger and Research Aircraft Measurements"

_EGUsphere, 2024_

## Referee Comment (RC1)

Discussion manuscript:

https://egusphere.copernicus.org/preprints/2024/egusphere-2024-2360/

The paper presents a mapping and adjustment procedure for IAGOS-MOZAIC and IAGOS-CORE humidity data from the ICH instruments using the more precise IAGOS-CARIBIC humidity measurements as reference. The idea to adjust ICH data looks certainly promising since it would allow to exploit the unique spatial and temporal coverage of the IAGOS-MOZAIC and CORE airborne data. As such, the resulting climatological data set can be of interest for a wider community of atmospheric scientists and is thus worth to be published. The paper is generally well written with a couple of minor issues stated in the specific comments and technical corrections. However, I have a couple of major concerns regarding the scope of the presented paper and the methodology which are detailed below and need to be addressed before publication.

**Major comments:**

- Scope of paper and journal: In the current state, the paper focusses on the description of mapping and adjustment methodology providing an adjusted H2O/RH data set as output. The scientific interpretation of the data set is limited to a short description of a few features which are directly evident from H2O maps at two different potential temperature levels (Fig. 10). In that sense, I see a lack of scientific novelty for an ACP research article and would rather shift it in the category "Technical note" where it would fit the scope of the journal. If the article shall stay in the research category, I strongly recommend to include one of the studies mentioned in the outlook section to strengthen the scientific impact (while dismissing parts which do not contribute to the main message of the paper, see below).

- Description of adjustment method (Sec 4.1): It is very hard for the reader to follow the description of the adjustment process since the paper only describes the performed steps without providing a context why they are performed. The authors should provide (1) a motivation why slicing the data as shown is done and (2) a structured guide through the process since the reader should theoretically be able to reproduce the data from the given description. Especially the reasoning behind the different FO thresholds and their subsequent treatment is not clear to me.

- In order to focus on the main message of the paper, I suggest moving the comparison between CARIBIC and JULIA to the appendix. It was certainly an important step in the data evaluation process, but a short description of the outcome (CARIBIC is suited as reference) is probably sufficient in the main text in order to focus on the main message of the work.

- A critical evaluation if adjusted data set has a real physical meaning in the LMS is missing in the paper. Or maybe more concise: Are you sure that ICH data in the LMS really represent the physical state of the atmosphere or could it just be "instrumental background"? At least from Fig 2 d and g this is very hard to tell and one might be trapped into fitting pretty constant background signals to also pretty constant stratospheric mean humidity values. Following that line of thought, is a bias correction for those altitudes justified given the very high variance in the data set? A discussion of detections limits, instrumental variance is necessary here.

- I would appreciate a quantitative discussion on the tradeoff "higher spatial-temporal coverage" vs "higher uncertainty" in Section 4.3 in order to see the added value of the adjusted MOZAIC/CORE dataset relative to the existing CARIBIC (optional JULIA) data sets.

- A statement on data availability is missing. Since the adjusted data set is the main result of this work, it should be available to the scientific community (at least after a certain amount of time). As a follow up on the previous comment, I would also encourage the authors to extend on the "guidelines" given in the outlook section regarding possible scientific use cases for the adjusted data set and state its limitations.

**Specific comments:**

- Fig. 3:
    - Dotted line is not explained in caption. Is it the actual flight path with color coded temperature?
    - Consequently, the bottom right RHliq Box in the figure should read $T^{(k+1)}$ instead of $T^{(k)}$.
- l. 161: The statement in that sentence is difficult to be confirmed by Fig 2: Where is the "wet stratospheric blop" from Fig 2f located in the potential temperature space?
- l. 272: from the Figure, there is no "scattering along the 1:1 line" below 6ppm but higher values for CARIBIC as also indicated by the reported mean value and stated in l. 279. Is the mean calculated for the whole 0..10ppm bin? Shouldn't it rather be a regression line with error margins?
- l. 284: Fig 4b does not show discrepancies, should it be Fig 4a?
- l. 286: that statement is strong and need discussion considering the difference lying at the edge of combined uncertainties and the error bars indicated in Fig 4a touching the identity line only for a rather small subset of points.
- l. 297ff: what is the hypothesis here? Is the fraction of RH values below 10% or the mean the driver for the bias as suggested by the reference? The pdfs in Fig 6b also show significantly higher values for MOZAI above 10% in both shown RH bins, so it looks more like a shift of the complete MOZAIC pdf to higher values plus a substantial increase in pdf width for the lower RH bin. I think that needs to be discussed since it is a clear indicator for the lower measurement limit of MOZAIC.
- l. 307: Fig 6a does not show distance to TP, please reformulate text.
- l. 314: obviously there are systematic biases, the filtering just eliminates one specific calibration issue -> please reformulate.
- l. 315: What is meant by "stronger variation of the bins"? is it higher variation of humidity values within the bins or a higher variation of bin mean values?
- Fig. 7a: black dot and black line is not explained, what are the leftover red lines? -> please consider revising the plot.
- l. 342: there are no regression lines in Fig 7cd -> do you mean the black dots?
- l. 359: Where is the black line in Fig 7b?
- l. 386: what is the "relative bias" and how is it calculated? Is that value considered as the uncertainty of the MOZAIC bin mean? In that case, this value is crucial for further handling of the data set and needs to be explained comprehensively. If I misinterpret that, such an uncertainty analysis has to be incorporated elsewhere in the manuscript.

- l. 391: due to which reason?
- Fig. 9a: the error budget seems very counter-intuitive: how can the error margin for the measurement uncertainty shrink to zero when approaching very low RH values? That looks like a flaw in the calculation method (or needs to be explained in detail)
- l.450: rephrase "detect" -> maybe "quantify"
- l. 452: There is likely also a sampling bias in IAGOS, e.g. due to avoidance of convective systems, turbulence and due to flying well within the jetstream on westbound flights and vice versa. That might lead to e.g. regions at the edge of the jetstream being underrepresented. Could be considered in the discussion.

**Technical corrections:**

l. 144: Figure 2 h & i instead of h-i

l. 154: Figure 2 e & f instead of e-f

l. 160: Figure 2f instead of Figure 2b

l. 163: Figure 2c instead of Figure 2a

l. 330: comma to be deleted

l. 316: right parenthesis missing

l. 359: calculated -> d is missing

l. 379: dot to be deleted

l. 457 & 458: $H_2O$ and $RH_{liq}$ with subscript

Fig. 8: reference in caption should be to Fig 6 instead of Fig 8

---

## Referee Comment (RC2)

**Review – Konjari et al., Water Vapor Climatologies in the UTLS…**

This paper provides a method for adjusting a long record – nearly thirty years! (1994-2022) – of water vapor measurements in the upper troposphere (UT) and lowermost stratosphere (LMS) acquired from sensors flown aboard commercial passenger aircraft. The adjustment of the long IAGOS-MOZAIC and -CORE (IMC) dataset, which utilized a compact capacitive humidity sensor (ICH), is achieved through a carefully constructed comparison with data from IAGOS-CARIBIC (ICB), which used a more sophisticated and sensitive frostpoint hygrometer (WaSul). The validity of the ICB measurements is separately verified via comparison with research quality data acquired by a Lyman-alpha hygrometer (FISH) compiled in the JULIA database. The purpose of the adjustment is to improve the accuracy of the long IMC record and to generate a more highly resolved climatology of water vapor in the Northern Hemisphere UT/LMS from this dataset suitable for scientific study and analysis.

The comparison and correction of the IMC dataset is complicated by the fact that the IMC and ICB data were acquired by sensors on different platforms flying at different times and locations. The methodology developed here attempts to correct for the impact of atmospheric variability by creating binned-average data differentiated by height above the tropopause, equivalent latitude, month, as well as concomitant temperature and ozone ranges, to facilitate comparison of water vapor measurements in dynamically and compositionally similar airmasses.

The goal of this work – to produce a high-resolution research-quality dataset of UT/LMS water vapor from the long record of IMC measurements – is ambitious, and the effort is substantial. However, the data mapping methodology is clever, defensible, and probably about the best that can be done. In the final section of the paper (4.3) prior to the Conclusions, the adjusted climatologies are evaluated. The spatial and temporal variability of water vapor in the resulting dataset is found to be consistent with expectations, indicating that the adjusted climatologies may be useful for further study. The paper constitutes an important contribution to the field, especially as it addresses the difficulties and limitations of measurements made on board passenger aircraft and presents a clever and well-reasoned attempt to overcome them. However, I have some significant reservations that should be addressed before I can recommend publication.

**Comments/Questions/Concerns:**

- The discussion of the actual adjustment procedure was difficult to follow. Please try to provide more clarity and detail.
- Is the adjusted dataset publicly available? I strongly believe it should be made available with a summary of the mapping methodology and a comprehensive list of caveats. See below…
- I find the statements regarding the "better resolution of temporal and spatial variability" to be misleading. The averaging in the adjustment process and derivation of the climatology limits the effective resolution of the product. In particular, the temporal resolution of this climatology is poor given that it relies on *multi-annual monthly mean values*. MLS, which provides daily, near-global profiles of water vapor from the UT to the mesosphere, has far superior temporal resolution, where monthly means from different years can be compared to examine inter-annual variability.
- Also, any assessment of trends in water vapor concentrations/changes in dynamical processes, etc. over this 30-year period is likely impossible given the limitations of the dataset. This constraint should also be emphasized.

**Minor comments/suggestions:**

Lines 8/9 – data set or dataset?

Line 8 – "applying" sounds a bit odd, maybe "utilizing" or "employing"?

Line 9 – add a space here "hygrometer (ICH)"

Line 26 – consider: "…and a corresponding **total stratospheric water vapor radiative feedback parameter** of 0.2 – 0.3 W/m2 per 1 Kelvin of surface warming." (?)

Line 30 – consider: "…UT/LMS H2O, as well as trends **at** high temporal…"

Line 38 – consider: "…passenger aircraft flights, **enabling the resolution of** strong…"

Line 48 – consider: "In this context, **the large quantity in-situ H2O measurements provided by IAGOS is important to improving the accuracy of**…"

Lines 52 and 53: spell out CORE and CARIBIC?

Line 55: "…measurements by a **compact capacitive humidity sensor**…" (replace all locations of humidity capacity sensor with capacitive humidity sensor)

Line 58: "…were found to **lose** precision…"

Line 63: "This lower detection limit for the ICH instrument **was later determined to be 30 ppmv** by means of a dedicated…"

Line 78: "…H2O variability in the UT/LMS at northern mid-latitudes."

Section 2.1: please spell out all acronyms?

Line 96: "…and **derived** H2O mixing ratios."

Line 110: spell out "PA" in "PA-laser spectrometer"

Line 126: "…ERA5 data **are used at reduced** resolution…"

Figure 2 caption: "…the probability density in coordinates relative…"

Line 142: "measurement quantity" here is confusing… I believe what is meant is "magnitude of RHliq measured by the ICH" and not the quantity of measurements. So, consider: "…closely linked to the magnitude of RHliq measured by the ICH sensor."

Lines 171+: consider: "We use a measurement mapping method to evaluate MOZAIC&CORE with respect to CARIBIC, focusing on the primary variable measured by MOZAIC&CORE, RHliq."

Line 173: remove space after the open parenthesis (500 flights)

Line 177: consider: "These factors particularly affect the UT…"

Line 186: consider: ", and the corresponding H2O mixing ratio."

Line 231: "This trade-off factor shows **that fewer measurements** are needed to constrain… providing **confidence in our approach**."

Line 241: "…serve as **a** reference…"

Line 288: what does the (2) refer to?

Line 292: "…for which this intercomparison is valid."

Figure 7 caption: "…the plots (c) and (d) show the derivation…" (delete "exemplary")

Line 329: "…consist of a large number of measurements **on** the order of…"

Line 330: "…state**.**" (delete comma)

Line 336: "For every one…" or "For each…"

Line 359: "…is calculated."

Section 4.2: There are a lot of separate paragraphs here, is that intentional?

Line 374: consider: "The adjusted MOZAIC&CORE-based H2O climatology offers the advantage of a longer record and greater spatial and seasonal sampling than the datasets of CARIBIC and JULIA, enabling more detailed analysis of the drivers of H2O variability. However, the adjustment of mean values requires… ."

Line 379: "." (?)

Line 383+: consider: "Despite the good agreement shown in Figure 8, the adjusted MOZAIC&CORE H2O incorporates uncertainties associated with (1) the measurement itself and (2) the adjustment methodology."

Line 387: "…is on the order of…" ("on" not "in")

Line 388: consider: "Uncertainties in the mapping method result from the small number of…"

Line 390: "…despite the fact that the sampling strategy should reduce the impact of uncertainties…"

Line 392: consider: "The uncertainty of the mapping method is determined as follows: The bias derivation… is performed for each season separately. In the next step, …from the four seasonal means. Finally, these standard deviations are used to do derive the mean standard deviation as a function solely of RHliq."

Figure 9 caption: (dashed line) not (solid line)

Line 397: "The uncertainty… **varies** depending on the.."

Line 407: clarification: "…same **geographic** latitudes."

Section 4.3: Again, there are a lot of separate two-sentence paragraphs here, is that intentional?

Line 409+: consider: "From fall to spring the highest values in the mid-latitudes… . Higher H2O amounts occur over the Atlantic than over continental regions during the winter half of the year, associated with greater low pressure activity over this area, and the resulting large scale uplift of moist and relatively warm airmasses (UT)… ."

Figure 10: Is the red box in panel (a) meant to highlight the isentropic mixing of moist air into the LS?

Line 415: "…Figure**s** 11a-d show adjusted multi-annual monthly means of adjusted H2O, plotted in coordinates of… "

Line 417: extra space before 100 ppmv?

Line 419+: The wording here is clunky and the discussion is perhaps too simple. In addition to the seasonal change in H2O along the 340 K potential temperature surface, there is a distinct shift in the tropopause height, such that 340 during the summer/fall it is near the tropopause level and during the winter/spring it is >20K above. Also, I see H2O for Jan in the range from 10-20 not 5-10? Perhaps I'm not reading the color bar correctly?

Line 423: mid-latitude not mid-latitudinal (?)

Figure 11 caption: extra space after H2O and before (e-h)

Line 432+: consider: "Finally, we examine how well sampled the UT/LMS is over the North Atlantic, given… "

Line 436: "Overall, good coverage is found."

Line 441: …adjust H2O climatologies (?) or H2O data (?)

Line 447: consider: "For the comparison, a mapping approach was utilized, where measurements were grouped into bins with similar dynamical origins and properties. Consideration of equivalent latitude, season, and height…, were used to derive mean RHliq values… ."

Line 455: "…showed good agreement in the…"

Line 456: distance to the tropopause is hard to parse here… consider: "However, in the LMS, the average values were generally biased, with the magnitude of the bias increasing with distance above the tropopause, reaching relative differences of 300% for H2O at around 5 ppmv."

---

## Author Response (AR1)

**Response Review 1**

(black: Comments by reviewer)
(red: Answers)
(blue: Text from the paper; bold: Changed / added text)

**Response to major comments**

(1) Scope of paper and journal: In the current state, the paper focusses on the description of mapping and adjustment methodology providing an adjusted H2O/RH data set as output. The scientific interpretation of the data set is limited to a short description of a few features which are directly evident from H2O maps at two different potential temperature levels (Fig. 10). In that sense, I see a lack of scientific novelty for an ACP research article and would rather shift it in the category "Technical note" where it would fit the scope of the journal. If the article shall stay in the research category, I strongly recommend to include one of the studies mentioned in the outlook section to strengthen the scientific impact (while dismissing parts which do not contribute to the main message of the paper, see below).

We agree that the focus of the paper is rather on the technical side. Therefore, the category will be changed to "Technical note".

(2) Description of adjustment method (Sec 4.1): It is very hard for the reader to follow the description of the adjustment process since the paper only describes the performed steps without providing a context why they are performed. The authors should provide (1) a motivation why slicing the data as shown is done and (2) a structured guide through the process since the reader should theoretically be able to reproduce the data from the given description. Especially the reasoning behind the different FO thresholds and their subsequent treatment is not clear to me.

We agree that in the discussion of the adjustment and the different steps of the procedure should be described with more clarity. We revised Ch. 4.1 (*see revised version; L.350 ff*) and also restructured Section 4.2 (*revised version; L. 436 ff*)

(3) In order to focus on the main message of the paper, I suggest moving the comparison between CARIBIC and JULIA to the appendix. It was certainly an important step in the data evaluation process, but a short description of the outcome (CARIBIC is suited as reference) is probably sufficient in the main text in order to focus on the main message of the work.

In our opinion, the evaluation of CARIBIC by JULIA should stay in the main paper. The FISH instrument is known to have a very high precision down to the lowest stratospheric mixing ratios, and there is no evaluation of the CARIBIC $H_2O$ measurements (WASUL + frost point hygrometer) so far. Furthermore, we wanted to provide some side notes on the differences between CARIBIC and JULIA concerning the respective "sampling strategies" (e.g. passenger aircraft vs flights to observe specific meteorological conditions), especially for people using one or both of these datasets for (climatological) studies.

[Figure]

**Figure B1**: Comprison of ICH, FISH, WASUL, and other hygrometer for a flight during the DENCHAR campaing. The light blue line indicates the saturation mixing ratio with respect to ice.
- Comment on WASUL: Not the same version as used in the CARIBIC package + additional use of a frost point hygrometer for CARIBIC.
- Comment on the moist bias of the ICH around 09:10 – 09:20: "Memory effect" due to the slow response time of the ICH for low temperatures; see response to Minor issues - (6)

Source: Rolf et al. (2024)

(4) A critical evaluation if adjusted data set has a real physical meaning in the LMS is missing in the paper. Or maybe more concise: Are you sure that ICH data in the LMS really represent the physical state of the atmosphere or could it just be "instrumental background"? At least from Fig 2 d and g this is very hard to tell and one might be trapped into fitting pretty constant background signals to also pretty constant stratospheric mean humidity values. Following that line of thought, is a bias correction for those altitudes justified given the very high variance in the data set? A discussion of detections limits, instrumental variance is necessary here.

It is known from previous studies that, despite that the ICH loses its sensitivity for low $RH_{liq}$ of 10% and less, there is still a response of the sensor, i.e. a bias as a function of $RH_{liq}$ (Neis et al., 2015; Rolf et al., 2014; see Figure B1) can be determined. This explains the clear relation that we found when comparing the ICH data to CARIBIC (Figure 5), which is indeed barely visible in Fig. 2 due to the scaling utilized for the low $RH_{liq}$ ranges. A complete loss of sensitivity occurs for low $RH_{liq}$ close to 1 %, which response to H2O of < 8 ppmv in the LMS (see Figure B2; compare saturation mixing ratio (with respect to ice, so even higher for liquid phase) with the measurements by FISH (dark blue line) ). Such low $RH_{liq}$ are, however, infrequent (see figure 6b; CARIBIC) at typical flight level height.

We agree that a more detailed discussion of the detection limit should be provided. A comment is added to Section 3 (Revised paper; *L. 144 ff*):

"Previous research campaign flights where a simple IAGOS ICH sensor was operated along with the sophisticated water vapor instrument FISH revealed that the sensitivity of the ICH sensor decreases significantly for very low $RH_{liq}$ values below 10 % (Neis et al., 2015a). The reason for this loss of sensitivity of the ICH sensor in the LMS is attributed to the adiabatic compression effect. As the air flows into the inlet towards the sensor, it undergoes heating in the range of 20 to 30 K. Consequently, even though the stratospheric humidity values are already very low, the measured values at the sensor become about 10 times lower or even less, resulting in a sensitivity loss of the ICH for these very small relative humidities. **Despite the systematic biases at low $RH_{liq}$, the ICH sensor demonstrates a measurable**

**response beyond the noise level, even at RH$_{liq}$ values as low as approximately 1% (Neis et al., 2015a; Rolf et al., 2023) .**

Because there is indeed a lower limit below which just instrumental background is measured, sampling bins consisting of such dry air masses should not be adjusted. In the datasets that will be published we therefore do not provided the adjusted data for bins with RH$_{liq}$< 4 %, which corresponds to RH$_{liq, adjusted}$ ~1-2 % (see Figure 5).

(5) I would appreciate a quantitative discussion on the tradeoff "higher spatial-temporal coverage" vs "higher uncertainty" in Section 4.3 in order to see the added value of the adjusted MOZAIC/CORE dataset relative to the existing CARIBIC (optional JULIA) data sets.

We added a comment in the text (*revised paper; L. 415 ff*):

"Multi-annual monthly means of adjusted H2O, based on all MOZAIC&CORE data, are shown in Figure 10 for two Θ levels, 335 K (a-d) and 350 K (e-h). The magenta solid line indicates the mean 2 PVU line. **The H$_2$O data is provided with a resolution of at least 5 ppmv. This relatively low resolution was chosen in regard of the uncertainties in the adjusted dataset. Specifically, for values below 20 ppmv, the uncertainty of the adjusted data can reach up to 30%. Therefore, a higher resolution would not be meaningful. Nonetheless, the given resolution is sufficient to capture spatial and seasonal features, also in areas where CARIBIC data is sparse or unavailable due to limited flight coverage (see discussion below)."**

(6) A statement on data availability is missing. Since the adjusted data set is the main result of this work, it should be available to the scientific community (at least after a certain amount of time). As a follow up on the previous comment, I would also encourage the authors to extend on the "guidelines" given in the outlook section regarding possible scientific use cases for the adjusted data set and state its limitations.

A dataset will be publicly available, and a doi will be provided within the paper as soon as the paper is accepted. This includes multi-annual climatologies for different regions (Atlantic, Europe, America, Asia) on Equivalent latitude – Relative to tropopause Coordinates (like Fig.11), and multi-annual means on different theta levels with a resolution of 5 x 5° (like Figure 10). Additionally, a bias estimate will be provided.

A summary of mapping approach as well as a list of caveats will be added to the dataset. Caveats that have to be emphasized are
○ limitations of the adjustment itself (e.g. uncertainties up to 30 %)
○ differences in the temporal coverage between different regions and resulting differences in the climatologies.

**Response to minor comments**

(1) Fig. 3:
- ○ Dotted line is not explained in caption. Is it the actual flight path with color coded temperature?

  Explanation added in caption.

- ○ Consequently, the bottom right RHliq Box in the figure should read T^(k+1) instead of T^(k).

  Revised.

(2) l. 161: The statement in that sentence is difficult to be confirmed by Fig 2: Where is the "wet stratospheric blop" from Fig 2f located in the potential temperature space?

Do you mean this statement (underlined)?: „This is evident, for instance, in a frequency of occurrence of $H_2O$ between 20 and 50 ppmv at distances of 3 km and more above the thermal tropopause (TTP) in Figure 2b, where the mean winter extratropical H2O values are expected to be below 10 ppmv on average (Zahn et al., 2014)"

- • For this reason, we cite a reference here, Zahn et al. (2014). In our paper, this is also evident from Figure 7.
- • In the potential temp. space, it occurs at around 350-360 K.

[Figure]

**Figure B2**: (a) Comparison of JULIA and CARIBIC H$_2$O; for JULIA the sampling was utilized for every campaign separately. Respective campaigns are indicated by different colors. (b) PDFs after excluding data from POLARCAT ('outlier filtered 'refers to the filtering of anomalously moist bins in JULIA). (c) MLS tape-recorder for the 380 K level.

(3)  l. 272: from the Figure, there is no "scattering along the 1:1 line" below 6ppm but higher values for CARIBIC as also indicated by the reported mean value and stated in l. 279. Is the mean calculated for the whole 0..10ppm bin? Shouldn't it rather be a regression line with error margins?

- Yes, the mean is calculated over the hole 0 – 10 ppm range, we clarified that in the revised version (*L. 267*): 'The comparison shows a scattering of the sampled mean H$_2$O values along an ideal regression line, but,  **averaged over the full 0 - 10 ppmv range**, a good agreement, …'

- We didn't draw a regression line because that would indicate a systematic bias of the CARIBIC data in the low mixing ratio range due to instrument limitations. However, there is the possibility that these low mixing ratios of less then ~ 6 ppmv might occur in the JULIA dataset more often because of a different sampling between passenger and research aircraft measurements.

  We further looked into the data to see from which campaign the different JULIA measurements come from. Figure B2 (a) shows the different sampling points in the comparison of JULIA and CARIBIC, with an additional sub-samping into the different campaigns (indicated by different colors). We found that the data at the driest end of

JULIA mostly come from one campaign, POLARCAT (dark green dots), which took place in 05-06/2008 mostly over Greenland. MLS also indicates anomalously dry stratospheric conditions during the campaign period, as shown for 380 K (Figure 2B (c); white box).

- We added more details in the revised paper (*L.278 ff*): 'Bins with JULIA indicating $H_2O$ of less than 3.5~ppmv are related to polar air masses **during one specific campaign, POLARCAT (...)**. It is not unlikely that these are cases of anomalously low mixing ratios that were purposefully measured during certain campaign flights and meteorological conditions, and are thus over-represented in JULIA.

  **When filtering out data from this campaign, the bias gets less pronounced but CARIBIC data are still moister by 0.5 to 1 ppmv for mixing ratios below 6 ppmv (not shown) which is outside the stated uncertainty of the JULIA and CARIBIC data (see Table 1).** It cannot be argued whether this small mean deviation is a result of different atmospheric sampling strategies between campaign and commercial aircraft flights, despite the strict filter conditions, or a small systematic bias.'

(4)    l. 284: Fig 4b does not show discrepancies, should it be Fig 4a?
       Yes, you are correct. Revised.

(5)    l. 286: that statement is strong and need discussion considering the difference lying at the edge of combined uncertainties and the error bars indicated in Fig 4a touching the identity line only for a rather small subset of points.

- We discussed this statement and concluded that it has to be reformulated. The discrepancies are partly outside the combined uncertainties which is mentioned in the revised text (see response to comment 3 – last point). We rephrased the last paragraph of Section 3.2.2 (*L. 281 ff*)

  'It cannot be argued whether this small mean deviation is a result of different atmospheric sampling strategies between campaign and commercial aircraft flights, despite the strict filter conditions, or a small systematic bias.  **Nevertheless, the possibility of a moist bias at the very dry end must be considered in the comparison between CARIBIC and MOZAIC&CORE, which is presented in the next section.**

[Figure]

[Figure]

**Figure B3:** (a) Schematic illustration of the measurement distribution (blue; blue line: average over all measurements) vs true state (red line). (b) $RH_{liq}$ measurements by the ICH (red) and FISH (black) during the CIRRUS-III campaign. The grey area indicates conditions during which the ICH has a slow reaction time. Figure by Neis et al. (2015).

(6)   l. 297ff: what is the hypothesis here? Is the fraction of RH values below 10% or the mean the driver for the bias as suggested by the reference? The pdfs in Fig 6b also show significantly higher values for MOZAIC above 10% in both shown RH bins, so it looks more like a shift of the complete MOZAIC pdf to higher values plus a substantial increase in pdf width for the lower RH bin. I think that needs to be discussed since it is a clear indicator for the lower measurement limit of MOZAIC.

- We created Figure 6a with the hypothesis that $RH_{liq}$ values below 10 % are biased, and therefore the driver of a bias of the mean values, as indicated by the reference.

- Figure 6b doesn't necessarily indicate that there is, on average, a bias also for $RH_{liq}>10$ % . The discrepancy between the PDFs for the $RH_{liq}>10$ % range can also be explained by the possibility that there is, for fixed $RH_{liq}$ ranges, not a more or less fixed bias but rather a broader distribution of measurements around a mean (biased) state, as illustrated in Figure B3 (a). Thus, a mean moist bias for a certain $\overline{RH_{liq}}$ increases the number of $RH_{liq}$ in the PDF also for higher values. We added more details in the revised paper (see L. 301 ff) and in the comment below

- Reason that lead to a pronounced scattering of the measurements around a mean state:

  - Uncertainties in the in-flight calibration (Revised paper; L.307 ff): '**Non-linear bias behavior could be attributed to uncertainties in the calibration the sensor's offset drift, which occurs between the routine ICH calibrations conducted every three months (Petzold et al., 2020). Although an in-flight calibration is performed to account for this sensor drift (Smit et al., 2008), uncertainties in the process (as noted in Smit et al. (2008)) can introduce non-linear variations in the bias of individual measurements between calibration intervals. Consequently, while this intercomparison study cannot determine the behavior of the bias for individual measurements, it does provide insights into the bias of mean values used in climatological studies.'**

  - Memory effect of the ICH: For low temperatures, the response time of the ICH is on the order of several 10th of seconds up to several minutes for temperatures close to 200 K (~230 K at sensor) . This is evident in Figure B3 (b) which shows the $RH_{liq}$ measurements by the ICH and the ones by FISH (stated as 'Reference (1 s)' ) during a CIRRUS-III field campaign flight (e.g. slow response time around 10:00 UTC).

- Overall, we can simply not say where the lower limit of MOZAIC occurs below which measurements are, on average, biased. This is misleading in the discussion of the plot, Figure 6a simply indicates that the (relative) bias of the MOZAIC bins gets more prominent the lower the $RH_{liq}$, below a certain threshold. We clarified that in the revised text (*L.293 ff*):

  In the comparison of MOZAIC to CARIBIC data, a distinct relationship is observed. When examining bins below the TTP where $RH_{liq}$ values are mostly above 30 %, MOZAIC and CARIBIC show a good agreement. This agreement is attributed to the infrequent occurrence  **of dry conditions with low and biased $RH_{liq}$.** Figure 6a illustrates the correlation between the relative number of MOZAIC measurements below 10 % $RH_{liq}$, which was stated Neis et al. (2015a) to be the upper limit for which ICH data show good quality, and the bias between MOZAIC and CARIBIC. Notably, the bias is prominent when the amount of measurements below 10 % $RH_{liq}$ is higher than 20 % (Figure 6a), while bins with fewer measurements below 10 % $RH_{liq}$ show little to no significant biases. **However, from this analysis it cannot be stated where the upper limit is situated above which the measurements are of good quality.**

(7)    l. 307: Fig 6a does not show distance to TP, please reformulate text.
   ○ Rephrased

(8)    l. 314: obviously there are systematic biases, the filtering just eliminates one specific calibration issue -> please reformulate.
   - Rephrased (L.: 'Although the resulting error was adjusted afterwards, we decided to utilize only data with the highest quality unaffected by this issue, to ensure that no systematic biases **due to the failure of the temperature sensor** affect the analysis.'

(9)    l. 315: What is meant by "stronger variation of the bins"? is it higher variation of humidity values within the bins or a higher variation of bin mean values?
   - Rephrased: 'However, for CORE, there is a stronger variation of the **bin mean values** compared to MOZAIC, …'

(10)   Fig. 7a: black dot and black line is not explained, what are the leftover red lines? -> please consider revising the plot.
   - The plot was revised and an explanation of the black line and dot was added to the Figure caption.

(11)   l. 342: there are no regression lines in Fig 7cd -> do you mean the black dots?
   - Yes, the black dots were meant. Rephrased.

(12)   l. 359: Where is the black line in Fig 7b?
   - Grey dots were meant. Rephrased.

(13) l. 386: what is the "relative bias" and how is it calculated? Is that value considered as the uncertainty of the MOZAIC bin mean? In that case, this value is crucial for further handling of the data set and needs to be explained comprehensively. If I misinterpret that, such an uncertainty analysis has to be incorporated elsewhere in the manuscript.

- The value is the uncertainty of the CARIBIC sampling bins. It is derived from the error propagation due to uncertainties in H2O, pressure and temperature:

$$RH_{liq} = \frac{e}{e_{sat}(T)} \quad with \quad e = p \cdot H_2O_{MR}$$

$$\Delta RH_{liq} = \frac{\partial RH_{liq}}{\partial H_2O_{MR}} \cdot \Delta H_2O_{MR} + \frac{\partial RH_{liq}}{\partial p} \cdot \Delta p + \frac{\partial RH_{liq}}{\partial T} \cdot \Delta T$$

$$= \frac{p}{e_{sat}(T)} \cdot \Delta H_2O_{MR} + \frac{H_2O_{MR}}{e_{sat}(T)} \cdot \Delta p + \frac{\partial e_{sat}}{\partial T} \cdot \frac{1}{(p \cdot H_2O_{MR})^2} \cdot \Delta T$$

- The uncertainty estimate of the CARIBIC sampling bins was incorporated in Section 3.3

(14) l. 391: due to which reason?

- Revised and further statement added:

'Furthermore, differences in the geographical coverage might also induce uncertainties, despite the fact that the applied sampling strategy should strongly decrease uncertainties in the comparison due to **differences in the geographical coverage. A further sub-sampling of the measurements into different geographical regions (N.Atlantic, N.America, Asia, Europe) didn't result in notable differences in the comparison of CARIBIC with MOZAIC&CORE.'**

(15) Fig. 9a: the error budget seems very counter-intuitive: how can the error margin for the measurement uncertainty shrink to zero when approaching very low RH values? That looks like a flaw in the calculation method (or needs to be explained in detail)

- That was my thought, too, when I first saw the plot. The reason for the shrinking of the uncertainty is that these low $RH_{liq}$ correlate with air masses at the upper part of the extratropical transition layer, where the natural variation of $H_2O$ is much lower. As a result, the uncertainty in the adjustment due to uncertainties in the sampling becomes very small for low $RH_{liq}$.

(16) l.450: rephrase "detect" -> maybe "quantify"

- Rephrased.

(17) l. 452: There is likely also a sampling bias in IAGOS, e.g. due to avoidance of convective systems, turbulence and due to flying well within the jetstream on westbound flights and vice versa. That might lead to e.g. regions at the edge of the jetstream being underrepresented. Could be considered in the discussion.

**Technical corrections:**

l. 144: Figure 2 h & i instead of h-i
l. 154: Figure 2 e & f instead of e-f
l. 160: Figure 2f instead of Figure 2b
l. 163: Figure 2c instead of Figure 2a
l. 330: comma to be deleted
l. 316: right parenthesis missing
l. 359: calculated -> d is missing
l. 379: dot to be deleted
l. 457 & 458: $H_2O$ and $RH_{liq}$ with subscript
Fig. 8: reference in caption should be to Fig 6 instead of Fig 8

All revised

**Response Review 2**

(black: Comments by reviewer)
(red: Answers)
(blue: Text from the paper; bold: Changed / Added text)

**Response to major comments**

(1) The discussion of the actual adjustment procedure was difficult to follow. Please try to provide more clarity and detail.

We agree that in the discussion of the adjustment, the different steps of the procedure should be described with more clarity. We revised Ch. 4.1 (see attached document).

(2) Is the adjusted dataset publicly available? I strongly believe it should be made available with a summary of the mapping methodology and a comprehensive list of caveats. See below…

A dataset will be publicly available, and a doi and a direct link to the data (iagos website) will be provided within the paper as soon as the paper is accepted. This includes multi-annual climatologies for different regions (Atlantic, Europe, America, Asia) on Equivalent latitude – Relative to tropopause Coordinates (like Fig.11), and multi-annual means on different theta levels with a resolution of 5 x 5° (like Figure 10). Additionally, a bias estimate will be provided.

A summary of mapping approach as well as a list of caveats will be added to the dataset. Caveats that have to be emphasized are
(1) limitations of the adjustment itself (e.g. uncertainties up to 30 %)
(2) differences in the temporal coverage between different regions and resulting differences in the climatologies

(3) I find the statements regarding the "better resolution of temporal and spatial variability" to be misleading. The averaging in the adjustment process and derivation of the climatology limits the effective resolution of the product. In particular, the temporal resolution of this climatology is poor given that it relies on multi-annual monthly mean values. MLS, which provides daily, near-global profiles of water vapor from the UT to the mesosphere, has far superior temporal resolution, where monthly means from different years can be compared to examine inter-annual variability.

The statement that the data provide "better resolution of temporal and spatial variability" (L.467) was indeed formulated in a misleading way. We wanted to emphasize that the adjusted IAGOS data provide an advantage compared to the CARIBIC dataset only, which includes a better spatial resolution being possible, due to a much better temporal resolution. The temporal resolution of the adjusted dataset itself however is very poor, given that multi-annual means are derived. We revised that statement in the paper.:

The adjusted climatologies provide better resolution of the  **seasonal** and spatial variability of UT/LMS H2O compared to other in-situ or space-borne dataset.

- Also, any assessment of trends in water vapor concentrations/changes in dynamical processes, etc. over this 30-year period is likely impossible given the limitations of the dataset. This constraint should also be emphasized.

That is an important point that should be emphasized.

Given the limitation, water vapor trends are not possible to asses, since the uncertainties of the adjusted data often exhibit the scale of possible trends. We added a comment on this issue in the revised manuscript (Section 4.2.2):

'Due to the requirement for a substantial amount of data and the relative uncertainty exceeding 10~\% in the driest range, robust trend analysis cannot be reliably performed using the derived data set. Even in regions with sufficient data availability, the level of uncertainty reflects the potential magnitude of H$_2$O trends.'

**Response to minor comments**

(green: corrected / considered)

- Lines 8/9 – data set or **dataset**?

- Line 8 – "applying" sounds a bit odd, maybe "utilizing" or "employing"?

- Line 9 – add a space here "hygrometer (ICH)"

- Line 26 – consider: "…and a corresponding total stratospheric water vapor radiative feedback parameter of 0.2 – 0.3 W/m2 per 1 Kelvin of surface warming." (?)

- Line 30 – consider: "…UT/LMS H2O, as well as trends at high temporal…"

- Line 38 – consider: "…passenger aircraft flights, enabling the resolution of strong…"

- Line 48 – consider: "In this context, the large quantity in-situ H2O measurements provided by IAGOS is important to improving the accuracy of…"

- Lines 52 and 53: spell out CORE and CARIBIC?

  Comment: No acronym for CORE

- Line 55: "…measurements by a compact capacitive humidity sensor…" (replace all locations of humidity capacity sensor with capacitive humidity sensor)

- Line 58: "…were found to lose precision…"

- Line 63: "This lower detection limit for the ICH instrument was later determined to be 30 ppmv by means of a dedicated…"

- Line 78: "…H2O variability in the UT/LMS at northern mid-latitudes."

- Section 2.1: please spell out all acronyms?

- Line 96: "…and derived H2O mixing ratios."

- Line 110: spell out "PA" in "PA-laser spectrometer"

- Line 126: "…ERA5 data are used at reduced resolution…"

- Figure 2 caption: "…the probability density in coordinates relative…"

- Line 142: "measurement quantity" here is confusing… I believe what is meant is "magnitude of RHliq measured by the ICH" and not the quantity of measurements. So, consider: "…closely linked to the magnitude of RHliq measured by the ICH sensor."

- Lines 171+: consider: "We use a measurement mapping method to evaluate MOZAIC&CORE with respect to CARIBIC, focusing on the primary variable measured by MOZAIC&CORE, RHliq."

- Line 173: remove space after the open parenthesis (500 flights)

- Line 177: consider: "These factors particularly affect the UT…"

- Line 186: consider: ", and the corresponding H2O mixing ratio."

- Line 231: "This trade-off factor shows that fewer measurements are needed to constrain…

- providing confidence in our approach."

- Line 241: "…serve as a reference…"

- Line 288: what does the (2) refer to?

  Comment: Equation 2; now clear in the text

- Line 292: "…for which this intercomparison is valid."

- Figure 7 caption: "…the plots (c) and (d) show the derivation…" (delete "exemplary")

- Line 329: "…consist of a large number of measurements on the order of…"

- Line 330: "…state." (delete comma)

- Line 336: "For every one…" or "For each…"

- Line 359: "…is calculated."

- Section 4.2: There are a lot of separate paragraphs here, is that intentional?

  Comment: We restructured Section 4.2

- Line 374: consider: "The adjusted MOZAIC&CORE-based $H_2O$ climatology offers the advantage of a longer record and greater spatial and seasonal sampling than the datasets of CARIBIC and JULIA, enabling more detailed analysis of the drivers of $H_2O$ variability. However, the adjustment of mean values requires… ."

- Line 379: "." (?)

- Line 383+: consider: "Despite the good agreement shown in Figure 8, the adjusted MOZAIC&CORE $H_2O$ incorporates uncertainties associated with (1) the measurement itself and (2) the adjustment methodology."

- Line 387: "…is on the order of…" ("on" not "in")

- Line 388: consider: "Uncertainties in the mapping method result from the small number of…"

- Line 390: "…despite the fact that the sampling strategy should reduce the impact of uncertainties…"

- Line 392: consider: "The uncertainty of the mapping method is determined as follows: The bias derivation… is performed for each season separately. In the next step, …from the four seasonal means. Finally, these standard deviations are used to do derive the mean standard deviation as a function solely of RHliq."

- Figure 9 caption: (dashed line) not (solid line)

- Line 397: "The uncertainty… varies depending on the.."

- Line 407: clarification: "…same geographic latitudes."

- Section 4.3: Again, there are a lot of separate two-sentence paragraphs here, is that intentional?

  Comment: We restructured the text.

- Line 409+: consider: "From fall to spring the highest values in the mid-latitudes… . Higher $H_2O$ amounts occur over the Atlantic than over continental regions during the winter half of the year, associated with greater low pressure activity over this area, and the resulting large scale uplift of moist and relatively warm airmasses (UT)… ."

- Figure 10: Is the red box in panel (a) meant to highlight the isentropic mixing of moist air into the LS?

    The red box highlights the area further studied in Figure 11. Description added in the capture.

- Line 415: "…Figures 11a-d show adjusted multi-annual monthly means of adjusted H2O, plotted in coordinates of… "

- Line 417: extra space before 100 ppmv?

- Line 419+: The wording here is clunky and the discussion is perhaps too simple. In addition to the seasonal change in H2O along the 340 K potential temperature surface, there is a distinct shift in the tropopause height, such that 340 during the summer/fall it is near the tropopause level and during the winter/spring it is >20K above. Also, I see H2O for Jan in the range from 10-20 not 5-10? Perhaps I'm not reading the color bar correctly?

    We added: '**This pattern can strongly be related to the increase of the tropopause Theta level during summer and the subsequently stronger influence of (isentropic) transport of H$_2$O from the subtropical regions into the mid-latitude LMS. Generally, layers in the LMS close to the TTP ($\Delta \Theta$ < 10 K) are moister during the summer season. A key question here is to what extent this increase can be attributed to local transport from the underlying upper troposphere (UT) or to large-scale transport, particularly from monsoon-influenced regions. Further trajectory-based analysis is essential to quantify the contributions of the different transport mechanisms involved.'**

- Line 423: mid-latitude not mid-latitudinal (?)

- Figure 11 caption: extra space after H2O and before (e-h)

- Line 432+: consider: "Finally, we examine how well sampled the UT/LMS is over the North Atlantic, given… "

- Line 436: "Overall, good coverage is found."

- Line 441: …adjust H2O climatologies (?) or H2O data (?)

    'Climatologies' might be a inaccurate, we wanted to make clear from the beginning of the summary that only mean values can be adjusted with our method. We rephrased the first sentence.

- Line 447: consider: "For the comparison, a mapping approach was utilized, where measurements were grouped into bins with similar dynamical origins and properties. Consideration of equivalent latitude, season, and height…, were used to derive mean RHliq values… ."

- Line 455: "…showed good agreement in the…"

- Line 456: distance to the tropopause is hard to parse here… consider: "However, in the LMS, the average values were generally biased, with the magnitude of the bias increasing with distance above the tropopause, reaching relative differences of 300% for H2O at around 5 ppmv."